 SciPost Phys. Lect.Notes 46 (2022)

# Introduction to Monte Carlo for matrix models

**Raghav G. Jha**

Perimeter Institute for Theoretical Physics, Waterloo, Ontario N2L 2Y5, Canada

raghav.govind.jha@gmail.com

## Abstract

We consider a wide range of matrix models and study them using the Monte Carlo technique in the large $N$ limit. The results we obtain agree with exact analytic expressions and recent numerical bootstrap methods for models with one and two matrices. We then present new results for several unsolved multi-matrix models where no other tool is yet available. In order to encourage an exchange of ideas between different numerical approaches to matrix models, we provide programs in PYTHON that can be easily modified to study potentials other than the ones considered here. These programs were tested on a laptop and took between a few minutes to several hours to finish depending on the model, $N$, and the required precision.

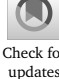

## 1   Introduction

The goal of this work is to introduce the numerical solution of matrix models in the large $N$ limit using Monte Carlo (MC) methods and implement them to solve several interesting multi-matrix models where no other analytical/numerical treatment is yet possible. The motivation for this work is partly from the recent progress in the numerical bootstrap program for matrix models and we hope that this introduction with PYTHON codes will assist those explorations and serve as a cross-check of the future results. Matrices play an important role and their presence is ubiquitous in many different areas ranging from nuclear physics to the study of random surfaces, conformal field theories, integrable systems, two-dimensional quantum gravity, and nonperturbative descriptions of string theory. It is well-known that several physical systems are explained to a great extent by normally distributed elements (Gaussian distribution). It is the most important probability distribution because it fits many natural phenomena. In this regard, Mark Kac[1] once made a remark which perfectly explains this - "That we are led here to the normal law (distribution), usually associated with random phenomena, is perhaps an indication that the deterministic and probabilistic points of view are not as irreconcilable as they may appear at first sight". The subject of random matrix theory is the study of matrices whose entries are random variables chosen from a well-defined distribution. It was Wishart who first noticed around 1928 that one can consider a family of probability distributions which is defined over symmetric, non-negative definite matrices sometimes also known as matrix-valued random variables now known as 'Wishart ensembles'. These are sometimes also known as 'Wishart-Laguerre' because the spectral properties of this distribution involve the use of Laguerre polynomials. But, the application of random matrices/distributions to physical problems was not until the 1950s when Wigner first applied the ideas of random matrix theory to understand the energy spectrum in nuclei of heavy elements. It was experimentally shown that unlike the case when the energy levels are assumed to be uncorrelated random numbers and the variable $s$ would be governed by the familiar Poisson distribution i.e., $P(s) = e^{-s}$, there was more to this story and the distribution was far from being like Poisson. He realized (what is now known by the name 'Wigner's surmise'[2]) that it could be described by a distribution given by $P(s) = \pi s/2 \, e^{-\pi s^2/4}$. The linear growth of $P(s)$

---

[1]Kac was a Polish American mathematician. His main interest was probability theory. He is also known apart from other things for his thought-provoking question - "Can one hear the shape of a drum?"

[2]Why is this called a 'surmise'? As is noted in the literature, the story goes like this: At some conference on Neutron Physics at the Oak Ridge National Laboratory in 1956, someone in the audience asked a question about the possible shape of distribution of the energy level spacings in a heavy nucleus. Wigner who was in the audience walked up to the blackboard and guessed the answer given above.

for small $s$ is due to quantum mechanical level repulsion (the fact that eigenvalues of random matrices don't like to stay too close) which was first considered by von Neumann and Wigner around 1930. This surmise and the paper written in 1951 [1] introduced the field of random matrix theory to nuclear physics and then in later decades to almost all of Physics.

This program was further continued in the 1960s when in their exploration of random matrices, Dyson and Mehta studied and classified three types (also called 'the threefold way') of matrix ensembles with different correlations. The first was 'Gaussian orthogonal ensemble' which was used to describe systems with time-reversal invariance and integer spin with weakest level repulsion between neighbouring levels and had $\beta = 1$. The second was the Gaussian unitary ensemble with no time-reversal invariance with intermediate level repulsion and $\beta = 2$. The third was the Gaussian symplectic ensemble for time-reversal invariance for half-integer spin with $\beta = 4$. These are now known as GOE, GUE, and GSE respectively.[3] The general $P(s)$ is given by, $c_\beta s^\beta e^{-a_\beta s^2}$ where $\beta \in (1, 2, 4)$ depending on the symmetry in question. For example, the nuclear data for heavy elements nearest neighbour spacing distribution is closely related to that of GOE distribution, see Fig. 2. The work of Dyson and Mehta made it more precise and improved it further compared to what is shown in Fig. 2 and explained in [2]. We note down $c_\beta$ and $a_\beta$ in Table (1) for three distributions and show them using MATHEMATICA in Fig. 1.

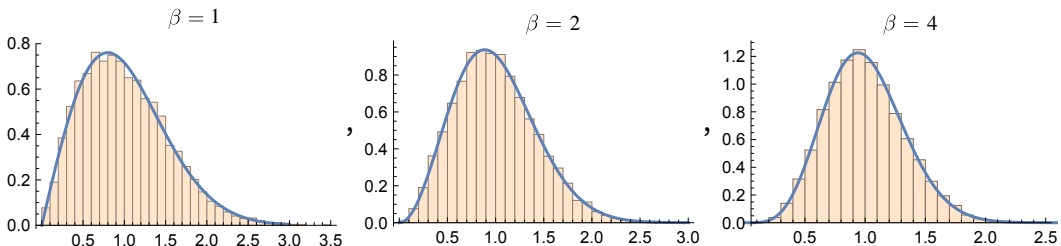

Figure 1: The distribution of three ensembles mentioned in the text.

Table 1: The constants $c_\beta$ and $a_\beta$ for different ensembles.

| $\beta$ | $c_\beta$ | $a_\beta$ |
|---|---|---|
| 1 | $\pi/2$ | $\pi/4$ |
| 2 | $32/\pi^2$ | $4/\pi$ |
| 4 | $2^{18}/3^6\pi^3$ | $64/9\pi$ |

It was later concluded much to Wigner's own surprise that his guess was fairly accurate as shown and improved by Mehta [3] and Gaudin [4]. Apart from its extensive use in Physics, the field of random matrix theory is intimately related to areas of Mathematics like number theory (especially the pair correlation of zeros of the Riemann-zeta function) and this was observed by Montgomery and Dyson in the 1970s. Some still believe that the secret of any future proof of the Riemann hypothesis lies in the deepest mysteries of random matrix theory. This belief is also the theme of the idea proposed independently by Hilbert and Pólya who suggested that the zeros of the zeta function might be the eigenvalues of some unknown Hermitian operator though no one knows about such an operator! We refer the reader to the excellent books [5,6]

---

[3]For example, GUE represents a statistical distribution over complex Hermitian matrices that have probability densities proportional to $\exp(-\text{Tr}(A^2/2\sigma^2))$ and where matrix elements i.e., $a_{ij}$ are an independent collection of complex variates whose real and imaginary parts are from a normal distribution with zero mean and unit variance. In MATHEMATICA , we can use: GaussianUnitaryMatrixDistribution[$\sigma$,N] to get such $N \times N$ matrix. We give sample code to get Wigner's famous semi-circle distribution in Appendix G.

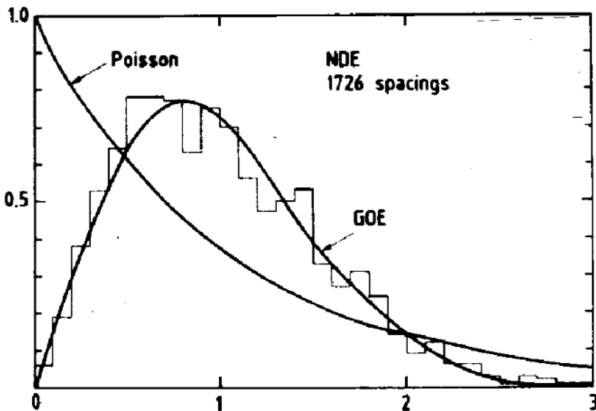

Figure 2: The nearest neighbour spacing distribution (i.e., $P(s)$) for nuclear data. The GOE (Gaussian Orthogonal Ensemble) and Poisson are shown by solid curves. This figure is taken from - 'Fluctuation Properties of Nuclear Energy Levels and Widths : Comparison of Theory with Experiment' by O. Bohigas, R. U. Haq, and A. Pandey.

for introductions to the field of random matrix theory.

The major development in the study of quantum field theories with matrix degrees of freedom satisfying some well-defined properties started with the work of 't Hooft in 1974 on the large $N$ limit of gauge theories. By then, it was accepted that the correct theory of strong interactions was QCD where we have matrix degrees of freedom for gauge fields based on $SU(3)$ gauge group. He proposed to consider a general $SU(N)$ symmetry with large $N$ and showed that in such a limit only planar diagrams survive and calculations become more tractable and the effects in QCD can then be explained as $1/N$ expansion. This work started an exposition of studying large $N$ limit of matrix valued fields and their applications to diverse areas of Physics and is extremely fruitful to date. This also enabled us to study several interesting features of quantum gravity from a field-theoretic point of view through the famous AdS/CFT conjecture. There are some excellent reviews about random matrix theory, matrix integrals/models, large $N$ limit and their formal aspects. We refer the reader to two excellent reviews written more than two decades apart [7,8] for detailed discussions.

We now outline the content of this article. In Sec. 2, we discuss saddle-point one-cut analysis of one matrix Hermitian model and provide a brief explanation about an alternative and equally effective method of orthogonal polynomials. We provide some additional details about the Ising model on a random graph which was solved using these polynomials in Appendix A. In Sec. 3, we start by mentioning the recent numerical bootstrap results for matrix models and then focus for bulk of the remaining section on explaining basics of the MC method and using it to solve different models. We discuss the solution of one-matrix model using MATHEMATICA in Appendix B and explain how to run PYTHON codes in Appendix C. Then we provide three PYTHON programs used in the paper in Appendix D, E, and F respectively. At last, in Appendix G, we give solutions to some exercises. All the programs can be accessed at:

https://github.com/rgjha/MMMC

## 2   Matrix models - Analytical and Bootstrap methods

Matrix models (or matrix integrals) are the simplest of models defined as integration over matrices in zero dimensions. In these cases, one evaluates integrals of the form:

$$Z = \int dM_i \cdots dM_j \exp\left[ -N\mathrm{Tr}\sum V(M_i) \right],$$
(1)

where $M_i$ are $N \times N$ matrices which can be Hermitian or unitary. In the study of zero-dimensional gauge theories, we encounter these types of matrix models where the integral is over some well-defined measure. These models often have interesting features in the large $N$ limit such as finite-volume phase transitions, volume reductions, factorization of correlation functions, etc. In the following subsection, we discuss solution of matrix models using two different methods in the planar limit.

### 2.1   Hermitian one-matrix model - Saddle point analysis

Before we delve into the details of how to solve the one matrix model using the saddle point (or 'stationary phase') method, we discuss the basics of this in a simple setting where we deal with integrals over a large number $N$ of variables. Suppose we want to evaluate the integral given by:

$$I(\alpha) = \lim_{\alpha \to 0} \int_{-\infty}^{\infty} e^{-\frac{1}{\alpha}f(x)}\, dx,$$
(2)

where $\alpha$ is a positive integer and $f(x)$ is a real-valued function. In the limit when $\alpha$ becomes small, the exponential causes the integrand to peak sharply at the function's minima. There might be several extrema, but the integral will be dominated by one which minimizes $f(x)$ as $\alpha \to 0$ (let it be $x_0$), we use Taylor expansion around the saddle point $x_0$ and ignore higher-order terms to get:

$$f(x) = f(x_0) + f''(x_0)(x - x_0)^2 + \cdots.$$
(3)

Using (3) in (2) along with the Gaussian integral[4] i.e., $\int_{-\infty}^{\infty} e^{-\alpha x^2} dx = \sqrt{\frac{\pi}{\alpha}}$, we get the desired result:

$$I(\alpha) = \sqrt{\frac{2\pi\alpha}{f''(x_0)}} e^{-f(x_0)/\alpha}.$$
(4)

> • Exercise 1: Find the terms which are of $\mathcal{O}(\alpha)$ in (4) and show that:
>
> $$I(\alpha) = \int_{-\infty}^{\infty} e^{-\frac{1}{\alpha}f(x)}\, dx = \sqrt{\frac{2\pi\alpha}{f''(x_0)}} e^{-f(x_0)/\alpha}\left[ 1 + \left[ \frac{5}{24}\frac{(f''')^2}{(f'')^3} - \frac{3}{24}\frac{f''''}{(f'')^2} \right]\alpha + \mathcal{O}(\alpha^2) \right].$$

We now move to the one-matrix model case where the role of $1/\alpha$ will be played by $N$ and hence in the planar limit, one can evaluate these integrals using the saddle-point method. This was first considered and famously solved by Brezin-Itzykson-Parisi-Zuber (BIPZ) [9]. This solution is standard and can be found in several reviews and textbooks, such as Ref. [7,10,11]. This model is solved using the method of resolvent and we will briefly sketch the solution

---

[4]It is claimed that Lord Kelvin once wrote $\int_{-\infty}^{\infty} e^{-x^2} dx = \sqrt{\pi}$ on the board and said -'A mathematician is someone to whom this is as obvious as that twice two makes four is to common man'.

described below. We start by writing $Z$ in terms of eigenvalues:

$$Z = \int dM \exp\left[-N \, \mathrm{Tr} V(M)\right] \tag{5}$$

$$= \int \prod d\lambda_i \Delta^2(\lambda) e^{-N \sum V(\lambda_i)}, \tag{6}$$

where $\Delta(\lambda) = \prod_{i>j}(\lambda_i - \lambda_j) = \exp\left[\sum_{i>j} \log|\lambda_i - \lambda_j|\right]$ is the Vandermonde determinant. If we vary one of the eigenvalues, it gives the saddle point equation:

$$\frac{2}{N}\sum_{j\neq i}\frac{1}{\lambda_i - \lambda_j} = V'(\lambda_i). \tag{7}$$

It is useful to introduce the density of eigenvalues,

$$\rho(\lambda) = \frac{1}{N}\sum_{i=1}^{N}\delta(\lambda - \lambda_i). \tag{8}$$

In the limit of large $N$, we can write (7) as:

$$V'(\lambda) = 2 \fint_b^a d\mu \frac{\rho(\mu)}{\lambda - \mu}, \tag{9}$$

where by $\fint$ we mean the Cauchy principal value of the integral. We often deal with symmetric single-cut such that $b = -a$. We can write resolvent by noting that it is the Stieljes transform of the eigenvalue density as:

$$G(z) = \fint d\mu \frac{\rho(\mu)}{\mu - z}, \tag{10}$$

which we can then write using Sokhotski-Plemelj theorem,

$$G(z \pm i\epsilon) = \fint d\mu \frac{\rho(\mu)}{\mu - z} \mp i\pi\rho(z), \tag{11}$$

and is equivalent using (9) to:

$$\lim_{\epsilon \to 0} G(z \pm i\epsilon) = -\frac{1}{2}V'(z) \mp i\pi\rho(z). \tag{12}$$

Once we find the resolvent, we can solve the model and find the moments through the eigenvalue density. It is useful to mention here that we can use the useful closed expression for resolvent in terms of a contour integral (see for example (A.24) of Ref. [12]) as given by:

$$G(x) = \int_{-a}^{a} \frac{-1}{2\pi i} \frac{\sqrt{x^2 - a^2}}{\sqrt{y^2 - a^2}} N V'(y) \frac{1}{x - y}, \tag{13}$$

for symmetric 'one-cut' solutions and by,

$$G(x) = \int_b^a \frac{-1}{2\pi i} \sqrt{\frac{(x-a)(x-b)}{(y-a)(y-b)}} N V'(y) \frac{1}{x - y} dy, \tag{14}$$

if the cut was instead $[b,a]$. For the case of 1MM with quartic potential i.e., $V(M) = \mu M^2/2 + g M^4/4$, one obtains the exact result (for $g \geq -\mu^2/12$):

$$t_2 = \frac{(12g + \mu^4)^{3/2} - 18\mu^2 g - \mu^6}{54g^2}. \tag{15}$$

It is straightforward to show that the end points of the cut is $[-a, a]$ with $a^2$ given by:

$$a^2 = \frac{2\mu}{3g}\left(\sqrt{1 + \frac{12g}{\mu^2}} - 1\right).$$

(16)

This one-cut solution is not valid for $g < -\mu^2/12$ and reduces to famous Wigner semi-circle law when $g \to 0$ with radius given by $2/\sqrt{\mu}$. The MATHEMATICA code to solve this model is given in Appendix B for the interested reader.

---

- Exercise 2: Show that $\det(V) = \prod_{i<j}(\lambda_i - \lambda_j)$ where $V$ is given by:

$$V = \begin{pmatrix} 1 & \lambda_1 & \lambda_1^2 & \cdots & \lambda_1^{N-1} \\ 1 & \lambda_2 & \lambda_2^2 & \cdots & \lambda_2^{N-1} \\ \vdots & \vdots & \ddots & \vdots & \vdots \\ 1 & \lambda_N & \lambda_N^2 & \cdots & \lambda_N^{N-1} \end{pmatrix} = \lambda_i^{j-1}.$$

---

- Exercise 3: Derive the loop equations given below:

$$\left\langle \mathrm{Tr}M^k V'(M) \right\rangle = \sum_{l=0}^{k-1}\langle \mathrm{Tr}M^l\rangle\langle \mathrm{Tr}M^{k-l-1}\rangle.$$

(17)

## 2.2 Method of orthogonal polynomials

In the last section, we discussed saddle-point methods to solve matrix models in the planar limit. But, this method is not very useful to study $1/N$ corrections. The preferred method for this purpose is based on orthogonal polynomials (OP). This was introduced by Bessis et al. in Ref. [13] and can be used to study matrix models with one and more matrices. The set of polynomials orthogonal with respect to the measure are defined as:

$$\int d\lambda e^{-V(\lambda)}P_n(\lambda)P_m(\lambda) = \int d\mu(\lambda)P_n(\lambda)P_m(\lambda) = a_n\delta_{mn},$$

(18)

where $d\mu(\lambda) = d\lambda e^{-V(\lambda)}$ is the measure. The basic idea is to rewrite the Vandermonde determinant appearing after we change from matrix basis to the basis of eigenvalues as,

$$\Delta(\lambda) = \det(\lambda_i^{j-1})_{1\leq i,j\leq N} = \det(P_{j-1}(\lambda_i))_{1\leq i,j\leq N}.$$

(19)

In order to illustrate how we can solve matrix models using this method, we use OP to study the Ising model on random graph in Appendix A. One can also use this method to study unitary matrix models like the famous one-plaquette model [14, 15] as was done in Ref. [16]. In addition to this model, there are other unitary models which are interesting for lattice gauge theory in lower dimensions. One such model is the external field problem which was considered in [17] and consists of a model of unitary links in external source field. It was solved with a fixed external field in the large $N$ limit and found to have a third-order phase transition. This model reduces to the GWW model when the external source is set to unit matrix. These developments in large $N$ QCD$_2$ models also lead (after a few years) to the idea of volume reduction in large $N$ limit known as Eguchi-Kawai reduction [18] in which it was shown that in the planar limit, the lattice gauge theory for an infinite lattice and unit cube are identical. The space-time seems incorporated in the large $N$ limit as an internal degree of freedom. There are some subtle requirements for this idea to work and are technical but this has led to lot of interesting work, see for instance Ref. [19].

## 2.3 Matrix bootstrap method

The basic idea of bootstrapping matrix models as proposed in Ref. [20] rests on the positivity (positive-definiteness) of the bootstrap matrix which we refer to as $\mathcal{M}$. For the case of one matrix model (1MM) with a potential $V(X)$ given by:

$$V(X) = \frac{1}{2}X^2 + \frac{g}{4}X^4, \tag{20}$$

the odd moments of $X$ vanishes i.e., $t_n = (1/N)\text{Tr}X^n = 0$ for odd $n$ while the even moments (of order greater than two) are all related to $t_2$. This renders the model simple to bootstrap since there is no growth of words (combination of matrix or matrices!) since all non-zero $t_n$ can be related to $t_2$.

> • Exercise 4: Use loop equations and show that for the 1MM with quartic potential, it is possible to write $t_4, t_6, t_8$ in terms of $t_2$. Also, check this either using MATHEMATICA or PYTHON [see Appendix for details]. Repeat this exercise for cubic potential where higher moments can be written in terms of $t_1$.

If we consider positive constraints that can be derived from $\langle \text{Tr}(\Phi^\dagger \Phi) \rangle \geq 0$ where $\Phi$ is a superposition of matrices which for one matrix model is $\Phi = \sum_k \alpha_k X^k$. This condition is equivalent to the positive definite nature of $\mathcal{M} \succeq 0$ where $\mathcal{M}_{ij} = \langle \text{Tr}X^{i+j} \rangle$. We can only enforce a subset of these constraints. For example, it was sufficient to access the positive definite nature of $\mathcal{M}_{6\times 6} \succeq 0$ and sub-matrices to get to the exact solution in Ref. [20]. For example, $\mathcal{M}_{2\times 2}$ is given by:

$$\mathcal{M}_{jk} = \begin{pmatrix} t_{2j} & t_{j+k} \\ t_{j+k} & t_{2k} \end{pmatrix} \succeq 0. \tag{21}$$

In this case, solving the model just means finding the bounds on $t_2$ since all others can then be calculated in terms of it (see the exercise above). Following this work, a quantum-mechanical model (in 0+1-dimensions) with up to two matrices was solved using similar techniques [21]. For the case of one-matrix model in this work, the Hamiltonian considered was given by:

$$H = \text{Tr}\left(P^2 + X^2 + \frac{g}{N}X^4\right), \tag{22}$$

and the authors considered trial operators up to length, $L = 4$ in this case and observed convergence to the expected result. Let us consider $L = 2$ as an example and consider operators like i.e., $\mathbb{I}, X, X^2$ and $P$. The bootstrap matrix of size $2^L \times 2^L$ which should be positive definite is constructed as:

$$\mathcal{M} = \begin{pmatrix} \langle \text{Tr}\mathbb{I} \rangle & \langle \text{Tr}X^2 \rangle & 0 & 0 \\ \langle \text{Tr}X^2 \rangle & \langle \text{Tr}X^4 \rangle & 0 & 0 \\ 0 & 0 & \langle \text{Tr}X^2 \rangle & \langle \text{Tr}XP \rangle \\ 0 & 0 & \langle \text{Tr}PX \rangle & \langle \text{Tr}P^2 \rangle \end{pmatrix} \succeq 0. \tag{23}$$

For the case of two-matrix quantum mechanics, it was observed that the convergence was slow but consistent with results expected using Monte Carlo results and bounds from the Born-Oppenheimer approximation method. Recently, another two-matrix integral given by the action (42) was recently solved in Ref. [22]. In this work, the authors used relaxation bootstrap methods (which takes us from non-linear semi-definite programming (SDP) problem to linear SDP) with $\Lambda = 11$ (which determines the size of the minor of the full bootstrap matrix) to constrain the moments of matrices such as $t_2 = \text{Tr}X^2/N$ and $t_4$ to six decimal places of precision for the symmetric case. We will come back to discuss this model for both symmetric and symmetry broken cases and its corresponding solution using MC methods in Sec. 3.3

and show that they are in perfect agreement. We hope that this yet small subsection on bootstrap methods would be extended in later versions of this article as more results accumulate in coming years.

# 3 Numerical solutions

There is only a selected list of analytically solvable matrix models in the planar limit. This inevitably brings the thought of attempting numerical solutions. Unfortunately, even here, there are not many methods that one can use. In fact, there are only two methods to our knowledge with the second method barely a few years old! This clearly signals the fact that there still remains a lot of work to be done in devising new numerical methods to solve matrix models. The most frequently used method is Monte Carlo (MC)[5] and it is quite effective but is not a panacea and has its own shortcomings. In this article, we will focus on the MC method while having explained the bootstrap solution in the previous section for the interested reader.

## 3.1 Basics of Monte Carlo method

The numerical method which is state-of-the-art in computations of matrix models and quantum field theories is the Monte Carlo approach. For higher-dimensional models, one starts with the lattice formulation which reduces the path-integral into many ordinary integrals. But even for a simple gauge theory like $\mathbb{Z}_2$ in four dimensions, this is not practical to evaluate. The fact that we need to do so many integrals suggests that may be some statistical interpretation and this is where the basic idea of Monte Carlo comes in. We make use of importance sampling (we sample states which are more relevant for the partition function more often compared to states which are not so relevant) in the Monte Carlo method. Using this sampling, one constructs a chain of configurations that approximately leads to the required distribution. Metropolis-Hastings algorithm and Hamiltonian/Hybrid Monte Carlo (HMC) are two frequently used methods that can generate a Markov chain using Monte Carlo (sometimes called Markov chain Monte Carlo) and lead to a unique stationary distribution. We will here focus on the latter since that has now become the preferred method in various numerical computations. This method was introduced in 1987 by Duane, Kennedy, Pendleton, and Roweth [23] who put together the ideas from Markov chain Monte Carlo (MCMC)[6] and molecular dynamics (MD) methods. Though we will mostly use HMC (discussed later) in this introduction, we present a simple example of Metropolis-Hastings (MH) algorithm to convince the reader that this sampling method indeed leads to desired results.

The MH algorithm is the most basic MCMC method for obtaining a sequence of random samples from a probability distribution for which direct sampling is difficult. This sequence can be used to approximate the distribution (histogram), or to compute the value of an integral. The MH algorithm works by simulating a Markov chain whose stationary distribution looks like the target distribution $\pi$ if sufficient time is allowed. This means that even though we don't apriori know the $\pi$, it will look like the samples are drawn from it in the long run.

---

[5]It is not widely known that Monte Carlo methods were central to the work required for the Manhattan Project and first introduced in the 1940s by Stanislaw Ulam and recognized first by von Neumann. He is often credited as the inventor of the modern version of the Markov Chain Monte Carlo method. In fact, the earliest use of MC goes back to the solution of the Buffon needle problem when Fermi used it in the 1930s but never published it. Since this work was part of the classified information, the work of von Neumann and Ulam required a code name. It was Metropolis who suggested using the name Monte Carlo based on the casino in Monaco where Ulam's uncle would borrow money from relatives to gamble. Ulam and Metropolis published the first paper on MC in September 1949 titled 'The Monte Carlo Method'.

[6]MCMC originated in the seminal paper of Metropolis et al. [24], where it was used to simulate the state distribution for a system of ideal molecules.

For complicated problems, however, it might be a very long time until we approach $\pi$. The algorithm was first introduced by Metropolis [24] and later generalized by Hastings [9] for asymmetric. As discussed, it draws sample from the proposal and accepts it with probability $\alpha$. If the sample is rejected the drawn sample remains unchanged otherwise it moves to a new configuration. The choice of good proposal distribution needs some thought, but a multivariate normal distribution usually does the required job. To implement MH algorithm, we must provide the 'transition kernel'. It is a simple update path of moving to a new position and we will denote this by $q(\mathbf{X}_t|\mathbf{Y})$. The steps of the algorithm are then summarized as follows:

- Choose $\mathbf{Y}$

- For $t = 1, 2 \cdots T$, we sample $\mathbf{X}_t$ from $q(\mathbf{X}_t|\mathbf{Y})$ where we have $\mathbf{Y} = \mathbf{X}_0$. Then we compute:

$$\alpha = \min.\left(1, \frac{\pi(\mathbf{X}_t)q(\mathbf{Y}|\mathbf{X}_t)}{\pi(\mathbf{Y})q(\mathbf{X}_t|\mathbf{Y})}\right) \tag{24}$$

and accept the new proposal with this probability.

If $q(\mathbf{X}_t|\mathbf{Y}) = q(\mathbf{Y}|\mathbf{X}_t)$ i.e., symmetric proposal then $\alpha$ simplifies and this was the original proposal by Metropolis. This was generalized to asymmetric proposals by Hastings [25]. We show results when this algorithm is applied to sample from distributions: $\pi(x) = \exp(-x), x \geq 0$ and $\pi(x) = \exp(-x^2)$ respectively. This is just an example since for these simple distributions better methods exist. We obtain the results shown in Fig. 3.

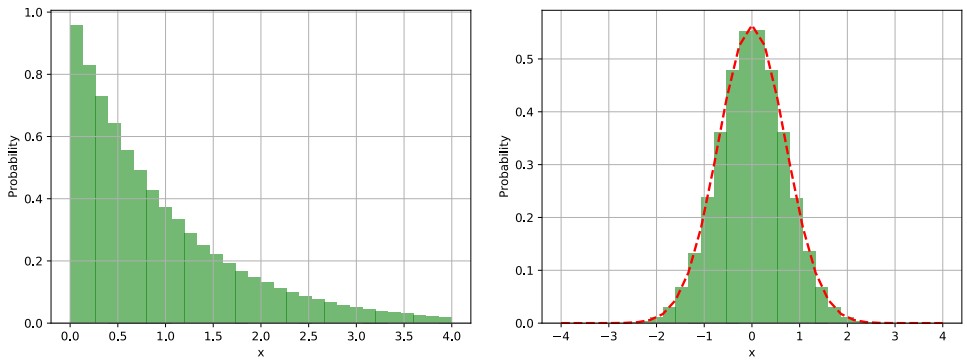

Figure 3: The application of MH algorithm to sample some simple target distributions. See text for details.

For a detailed review about HMC and its extension to rational HMC (RHMC) which is required for fermions, the interested readers can consult Ref. [26, 27]. The two basic parts of HMC [7] are, a) Use of integrator to evolve and propose a new configuration, b) accept or reject the proposed configuration. But before we discuss HMC, it is important to see how we generate random momentum matrices for the leapfrog integrator at the start of each trajectory (time unit) since this is necessary to ensure that we converge to the correct answer using Monte Carlo methods.

### 3.1.1 Random generator - Box-Muller algorithm

It is essential during HMC algorithm that we correctly generate random $N \times N$ momentum matrices at the start of the leapfrog method whose elements are taken from a Gaussian distribution. In this part, we will explain this procedure. The part of code that implements this can

---

[7]For the bosonic fields as is the case in this article, it is certainly possible to also use heat-bath algorithm but we introduce HMC since it is more natural when advancing to field theories with fermions where RHMC is extensively used

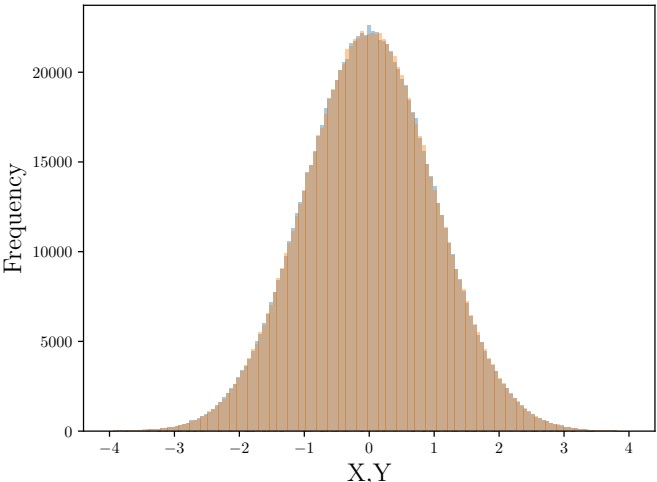

Figure 4: We show that in the limit of large sample size (here $10^6$), the prescription we use for generating random numbers tends to the desired Gaussian distribution with mean zero and unit variance.

be found in one of the sub-routines in Appendix D. Suppose we have two numbers $U$ and $V$ taken from uniform distribution i.e., (0,1) and we want two random numbers with probability density function $p(X)$ and $p(Y)$ given by:

$$p(X) = \frac{1}{\sqrt{2\pi}} e^{-X^2/2}, \tag{25}$$

and,

$$p(Y) = \frac{1}{\sqrt{2\pi}} e^{-Y^2/2}. \tag{26}$$

Since $X$ and $Y$ are independent, we can write:

$$p(X,Y) = p(X)p(Y) = \frac{1}{2\pi} e^{-R^2/2} = p(R,\Theta), \tag{27}$$

where $R = X^2 + Y^2$. This allows us to make the following identification:

$$U = \frac{\Theta}{2\pi}, \tag{28}$$

and,

$$V = e^{-R^2/2} \implies R = \sqrt{-2\log(V)}. \tag{29}$$

This implies,

$$X = R\cos\Theta = \sqrt{-2\log(V)}\cos(2\pi U), \tag{30}$$

$$Y = R\sin\Theta = \sqrt{-2\log(V)}\sin(2\pi U). \tag{31}$$

It is straightforward to check that it indeed generates a Gaussian distribution with desired properties and is shown in Fig. 4.

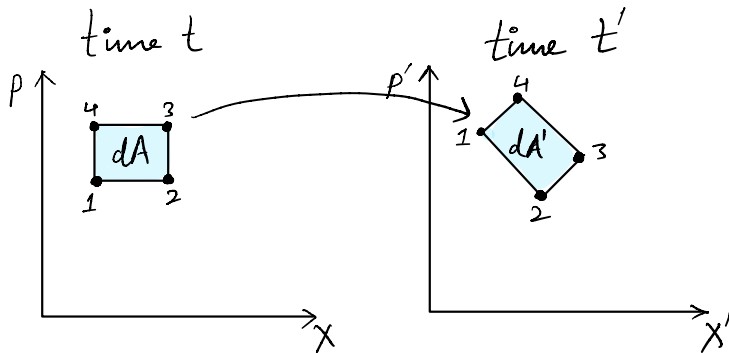

Figure 5: A representation of conservation of phase space area under the evolution of the system. See text for details.

### 3.1.2 Leapfrog integrator and accept/reject step

The leapfrog method is used to numerically integrate differential equations. This is a second-order method and the energy non-conservation depends on the square of step-size used. This integrator is *symplectic*, i.e., it preserves the area of the phase space. We can understand this as follows: Consider a region of area $dA$ as shown below in Fig. 5. The four corners at time $t$ are denoted by $(x, p), (x+dx, p), (x+dx, p+dp), (x, p+dp)$. At some later time $t'$, this will change to form corners of some other quadrilateral as shown with area $dA'$. It is then the statement of Liouville's theorem[8] that the areas are equal, i.e., $dA = dA'$. Using this idea we can easily prove important equality used to check MC computations employing symplectic integrators. Another important property that must be satisfied by our integrator is *reversibility*. Suppose we start with a field configuration $X$ and momentum taken from Gaussian distribution $P$ and evolve this for some time $t$ to a new set of field and momentum i.e., $X_1, P_1$. If we now reverse the momentum sign and evolve $X_1, -P_1$ for the same time $t$, then we will end up at $X_2, P_2 = (X, P)$. The reversibility ensures that our implementation will have the desired stationary distribution. It is left as an exercise for the reader to check that this is true in our programs. There are other integrators that are more efficient and lead to improvement such as Omelyan integrator but for the purpose of the models we study, leapfrog is more than sufficient. Let us now describe this algorithm satisfying these properties. We will use the following definition of forces:

$$f_i = -\frac{\partial S}{\partial X_i} = -N \frac{\partial \operatorname{Tr} V}{\partial X_i} \ , \quad P_i = \frac{\partial X_i}{\partial \tau} \tag{32}$$

where $N$ is the size of the matrix, $V$ is the potential of the model, $i$ is the number of matrices in the model, and $X_i$ are the Hermitian matrices. The basic steps of the leapfrog algorithm is given as:

- $X_i\left(\frac{\Delta\tau}{2}\right) = X_i(0) + P_i(0)\frac{\Delta\tau}{2}$

- Now several inner steps where $n = 1 \cdots (N'-1)$

  $P_i(n\Delta\tau) = P((n-1)\Delta\tau) + f_i((n-\frac{1}{2})\Delta\tau)\Delta\tau$

  $X_i\left(\left(n+\frac{1}{2}\right)\Delta\tau\right) = X_i\left(\left(n-\frac{1}{2}\right)\Delta\tau\right) + P_i(n\Delta\tau)\Delta\tau$

---

[8]Note that Liouville theorem is closely related to detailed balance condition which says that in equilibrium there is a balance between any two pairs of states i.e., equal probability.

- $P_i(N'\Delta\tau) = P_i((N'-1)\Delta\tau) + f_i((N'-\frac{1}{2})\Delta\tau)\Delta\tau$

- $X_i(N'\Delta\tau) = X_i\left(\left(N'-\frac{1}{2}\right)\Delta\tau\right) + P_i(N'\Delta\tau)\frac{\Delta\tau}{2}$

---

• Exercise 5: Show that a consequence of phase space conservation i.e., $dPdX = dP'dX'$ is that $\langle e^{-\Delta H(\cdot)}\rangle = 1$, where $\cdot$ denote the fields and $\Delta H = H'-H$. Using Jensen's inequality this implies that $\langle H'-H\rangle \geq 0$. Check this holds in a given MC evolution of multi-matrix model within errors after ignoring sufficient data for thermalization cut.

---

In the last step of HMC, a Metropolis test is carried out to accept or reject the proposed configuration. Suppose we start from the configuration $X$ of one-matrix model which is a $N \times N$ matrix and carry out the leapfrog part with some parameters and obtain a new configuration $X'$. The test then computes $\texttt{min.}(1, e^{-\Delta H})$ and generates a uniform random number between $r \in [0, 1]$. The new configuration is rejected if $\texttt{min.}(1, e^{-\Delta H}) < r$ otherwise it is accepted. We return to old fields if rejected and repeat this process.

### 3.1.3 Autocorrelation and error estimation

It must be kept in mind that given a Markov chain, the new states (i.e., configurations) can be highly correlated to previous ones. In order to ascertain that the measurement of the expectation value of an observable $\langle\mathcal{O}\rangle$ is not affected by correlated configurations, it is essential for proper statistical analysis to know the extent to which they are correlated. In this regard, it is important to measure the autocorrelation time $\tau_{\text{auto}}$ which measures the time it takes for two measurements to be considered independent of each other. So, if we generate $L$ configurations, then actually only $L/\tau_{\text{auto}}$ are useful for computing averages. We define the autocorrelation function of observable $\mathcal{O}$ such that $C(0) = 1$ as defined below:

$$C(t) = \frac{\langle\mathcal{O}(t_0)\mathcal{O}(t_0+t)\rangle - \langle\mathcal{O}(t_0)\rangle\langle\mathcal{O}(t_0+t)\rangle}{\langle\mathcal{O}^2(t_0)\rangle - \langle\mathcal{O}(t_0)\rangle^2} \, . \tag{33}$$

The behaviour of $C(t)$ is $\sim \exp(-t/\tau_{\text{auto}})$ for large $t$. This is called exponential autocorrelation time. We can also compute the 'integrated autocorrelation time' defined as:

$$\tau_{\text{auto}}^{\text{int.}} = \frac{\sum_{t=1}^{\infty}\langle\mathcal{O}(t_0)\mathcal{O}(t_0+t)\rangle - \langle\mathcal{O}\rangle^2}{\langle\mathcal{O}^2\rangle - \langle\mathcal{O}\rangle^2} \, . \tag{34}$$

We can write this in terms of a sum over autocorrelation function as: $\tau_{\text{auto}}^{\text{int.}} = 1 + \sum_{t=1}^{N} C(t)$. In general, $\tau_{\text{auto}}$ increases with system size, close to the critical point. One can express the statistical error in the average of $\mathcal{O}$ denoted by $\delta\mathcal{O}$ is given in terms of variance and integrated autocorrelation time as:

$$\delta\mathcal{O} = \sigma\sqrt{\frac{2\tau_{\text{auto}}^{\text{int.}}}{N}} \, , \tag{35}$$

where we have usual definitions i.e., $\sigma = \sqrt{\langle\mathcal{O}^2\rangle - \langle\mathcal{O}\rangle^2}$ and $N$ is the number of measurements. We now turn to the estimation of the errors. For this purpose, we use the Jackknife method which is also known as 'leave-one' method. We first split the data into $M$ blocks, with block size more than the autocorrelation time. In case, the data has no correlations, block length can be set to unity. The general procedure begins with discarding one block and calculating errors on the remaining ones. This is done for all $M$ blocks and error is estimated. The PYTHON code to perform this error analysis is given for the interested reader in Appendix F.

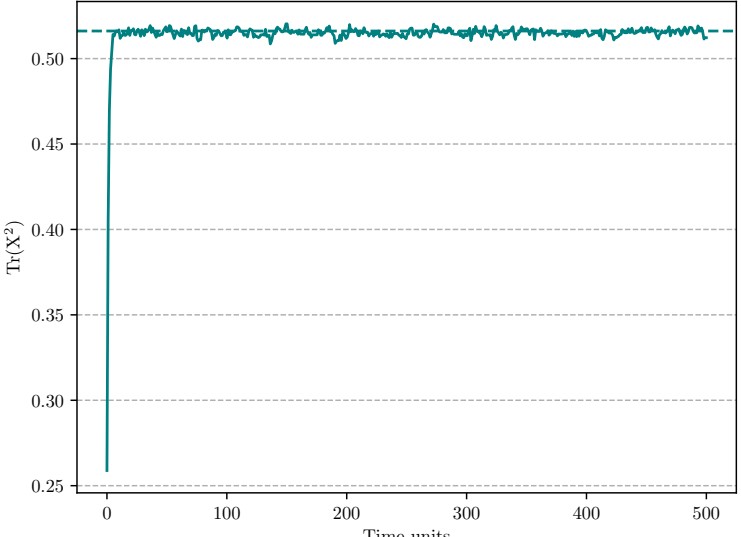

Figure 6: The computed value of $\mathrm{Tr}(X^2)$ with MC methods for quartic potential one matrix model is consistent with that obtained using analytical saddle-point methods (shown by dashed lines). These results are with $N = 300$ and $g = 1$ and took about 40 minutes on a laptop.

## 3.2 One-matrix model: Confirming exact results

In the previous subsection, we provided a very quick introduction of the basic elements of the MC method. We will now use it to study matrix models at large $N$. Before we embark on more complicated models, we should cross-check with known results. For this purpose, the exact solution available for one-matrix model is a good testbed. We start with the quartic potential given by:

$$V(M) = \frac{M^2}{2} + \frac{gM^4}{4}. \tag{36}$$

This has an exact solution and all moments, $t_n$, can be obtained using the MATHEMATICA code given in Appendix B. We show that the exact result and the Monte Carlo results agree for $g = 1$ in Fig. 6. The PYTHON code which was used to study this model can be found in Appendix D. We discuss how to run the code on your laptop and the estimated time to completion in Appendix C. The latest version of this code[9] is available at:

https://github.com/rgjha/MMMC/1MM.py

The quartic potential one-matrix model has a well-known critical coupling i.e., $g_c = -1/12$ below which solutions cease to exist. One natural question that comes to mind is: How well can MC capture this critical coupling?. We explored this question with our MC code and find that it is very effective in detecting the critical $g$ for this one matrix model and other multi-matrix models. For example, we obtained correct results for $t_2$ given by (15) until $g_{\mathrm{min.}} \sim -0.0819$ with $N = 300$ but the numerics did not converge (becomes unstable!) for $g = -0.0820$. It took about 1-2 hours of computer time to locate the critical coupling to accuracy of about $\sim 0.0014$ (i.e., $g_{\mathrm{MC}} - g_c \sim 0.0014$). We can always increase $N$ to get a more precise determination of the critical coupling.

---

[9]Please email the author for any bug report or additional requests

We can also consider one-matrix model with cubic interaction instead of quartic potential as above. The potential is given by:

$$V(M) = \frac{M^2}{2} + \frac{g_3 M^3}{3}. \tag{37}$$

It is well-defined only when $g_3 \leq 0.21935$. This is the radius of convergence of the planar perturbation series. Though this can be exactly solved, we encourage the reader to attempt Exercise 6 to numerically solve this using MC. This exercise provides good practice on how we can modify the potential in the given codes to study another model of interest.

> • Exercise 6: Check that one-matrix model PYTHON program given in Appendix D reproduces the correct result for the cubic potential i.e., $V(M) = M^2/2 + g_3 M^3/3$.

Another interesting matrix model closely related to the one matrix model was studied in the context of understanding the Yang-Lee edge singularity [28] in Ising model on random graphs. It is given by:

$$Z = \int \mathcal{D}X \mathcal{D}M \exp N \operatorname{Tr} \left( -\frac{X^2}{2} + \frac{gX^4}{4} - \frac{M^2}{2} + g\sqrt{\zeta} M X^3 \right). \tag{38}$$

After integrating out one of the matrices, $M$, we can reduce it the familiar one-matrix model problem,

$$Z = \int \mathcal{D}X \exp N \operatorname{Tr} \left( -\frac{X^2}{2} + \frac{gX^4}{4} + g^2 \zeta \frac{X^6}{2} \right). \tag{39}$$

Note that if we set $\zeta = 0$, this reduces to the 1MM problem but with negative sign for the quadratic term different from the positive sign we studied for one matrix model above. However, this is also exactly solvable and left as an exercise.

> • Exercise 7: Check that (39) follows from (38) and modify the potential for the 1MM PYTHON code to study this model.

### 3.3 Hoppe-type matrix models: Confirming bootstrap results

We now turn our attention to matrix models with a commutator interaction term. To our knowledge, this model was first introduced by Hoppe [29] and solved later by different methods in Refs. [30,31]. The partition function is given by:

$$Z = \int \mathcal{D}X \mathcal{D}Y \exp \left[ -N \operatorname{Tr}(X^2 + Y^2 - h^2 [X,Y]^2) \right]. \tag{40}$$

At large values of commutator coupling i.e., $h \to \infty$, this model becomes commuting with $[X,Y] \to 0$. The presence of commutator term in matrix models is common especially in models which have a dual gravity interpretation of emergent geometry behaviour. The exact result for average action is:

$$2\langle S_c \rangle + \langle S_q \rangle = N^2 - 1, \tag{41}$$

where $S_c = -Nh^2 \operatorname{Tr}[X,Y]^2$ and $S_q = N \operatorname{Tr}(X^2 + Y^2)$. This average action serves as a good check of the code. We can alternatively also consider a slightly more general two-matrix model which reduces to Hoppe's model mentioned above in a special limit. Such a matrix model is generally not solvable. This model was considered in Ref. [22] and solved using bootstrap methods and is given by:

$$Z = \int \mathcal{D}X \mathcal{D}Y \exp \left[ -N \operatorname{Tr}(X^2 + Y^2 - h^2 [X,Y]^2 + gX^4 + gY^4) \right]. \tag{42}$$

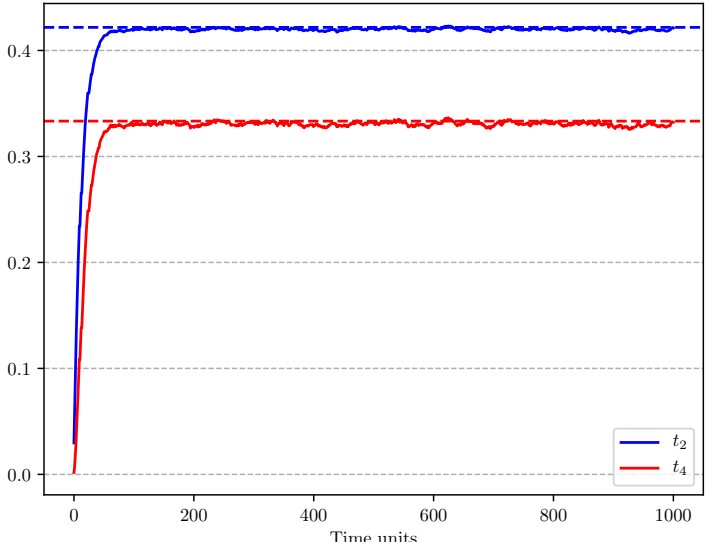

Figure 7: The matrix model defined by (42) is not solvable for generic $g$ and $h$ and was recently studied using bootstrap methods. We show the MC results by solid lines and those obtained using bootstrap by dashed lines. We get few digits of accuracy with 1000 time units by running for about 1.5 hours on a laptop. The results shown are with $g = h = 1$ and $N = 300$. For larger $N = 800$, the same run will take about 16-18 hours. We did an extended run for about 80-85 hours accumulating 5000 time units and obtained $t_2 = 0.42179(3)$ and $t_4 = 0.33336(5)$ with $N = 800$. The bootstrap results are $0.421783612 \leq t_2 \leq 0.421784687$ and $0.333341358 \leq t_4 \leq 0.333342131$ [22]. In fact, we can easily compute higher moments and we find that $t_{16} = 0.7153(8)$ and $t_{32} = 6.96(8)$.

The action in (42) has $\mathbb{Z}_2^{\otimes 3}$ symmetry (i.e., $X \to Y$, $X \to -X$, and $Y \to -Y$). For $g = 0$, it can be reduced to a matrix model which can be solved via saddle-point analysis or through the reduction to a Kadomtsev-Petviashvili (KP) type equation. For $h = 0$ it reduces to two decoupled one-matrix models and for $h = \infty$ as mentioned above, we have $[X, Y] = 0$ and it becomes similar to an eigenvalue problem. We considered this model with $g = h = 1$ using MC methods and show in Fig. 7 that the results obtained are consistent with Ref. [22]. This result is for $N = 300$ and took about $5500$ seconds on a 2.4 GHz i5 A1989 MacBook Pro. We also show the eigenvalue distribution for this case in the left panel of Fig. 8. The code for this model is available at:

https://github.com/rgjha/MMMC/2MM.py

We now consider the case where $g = 0$ and $h = h_c = -0.04965775$, the bootstrap results were not very accurate because of slow convergence. We obtained MC results for this case and obtain: $t_2 = 1.1886(15)$ and $t_4 = 2.866(20)$ with $N = 300$ after about 3-4 hours of run on a laptop. We also explored $h < h_c$ and find that MC breaks down and moments quickly runs away to infinity signalling that the model is not well-defined. We show the eigenvalue distribution for this case in the right panel of Fig. 8. It is interesting to consider the same model by flipping the sigs of the quadratic terms in $X, Y$ i.e.,

$$Z = \int \mathcal{D}X\mathcal{D}Y \exp\left[-N \operatorname{Tr}(-X^2 - Y^2 - h^2[X, Y]^2 + gX^4 + gY^4)\right]. \tag{43}$$

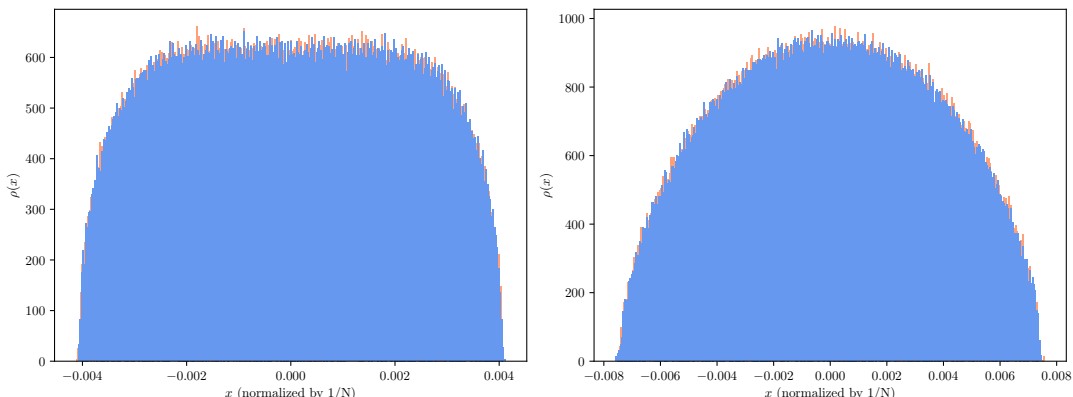

Figure 8: Left: The eigenvalue distribution of two matrices for $g = h = 1$ with $N = 300$. Right: The parabolic distribution for $g = 0, h = -0.04965775$ with $N = 300$.

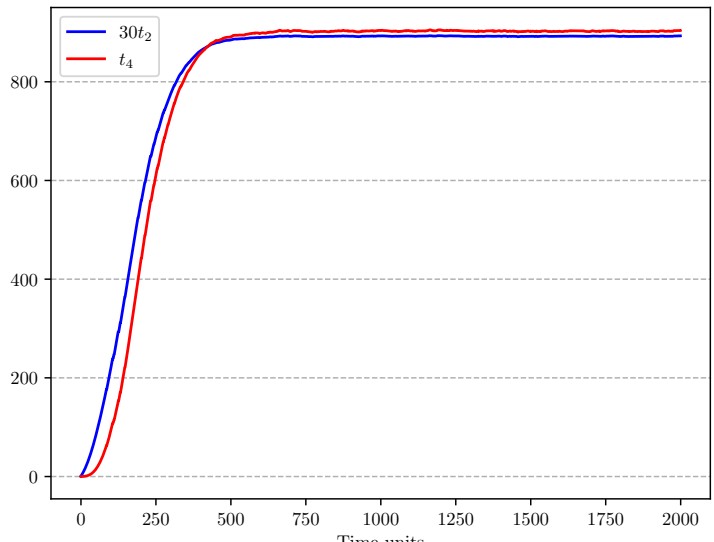

Figure 9: We show $t_2$ and $t_4$ obtained from MC. The values we obtain for this specific stream of run is for $N = 300$ with $t_2 = 29.73(3)$ and $t_4 = 903(3)$. This took about 3.5 hours on a laptop since it thermalizes late compared to symmetric case and we need to run for a longer time. This specific dataset corresponds to the red point in Fig. 10.

This corresponds to breaking the $\mathbb{Z}_2^{\otimes 2}$ symmetry and just keeping the $X \rightarrow Y$ symmetry. This model was studied using bootstrap methods in Ref. [22] but compared to the symmetric case, it was tough to get the same level of accuracy in the bootstrap results for this case. To understand how well MC works for this case, we explored this and see good agreement where applicable. The results are shown in Fig. 10. The bootstrap method has the advantage that the entire boundary line can be obtained at once while we need to do multiple streams of runs in Monte Carlo to get all points. It seems likely that one can produce the entire set of solutions by starting from different initial matrices at the start of MC process. It will be interesting to apply any other method in the future to this case and see how it performs against MC and

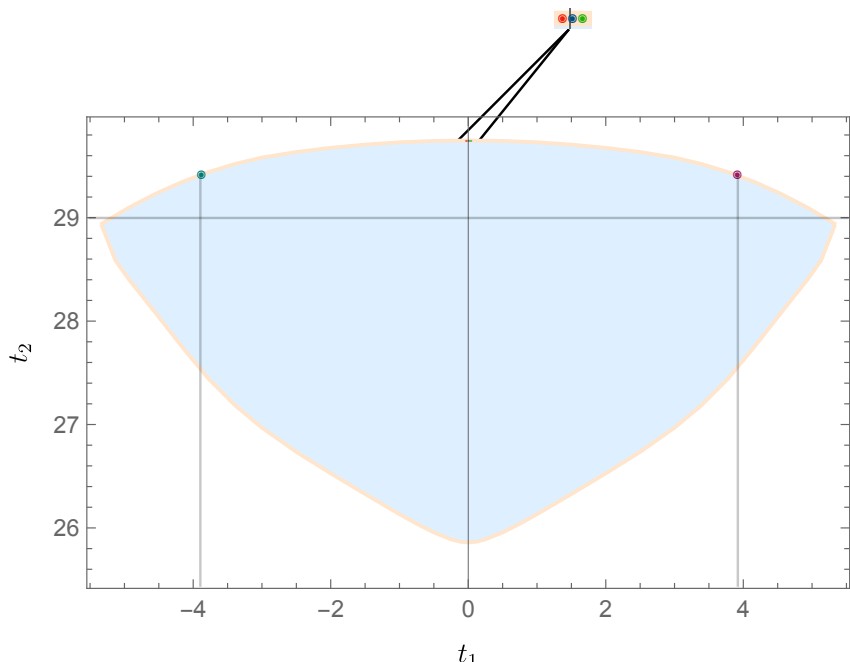

Figure 10: We show that the MC results from different starts give different results for $t_1$ and $t_2$. This is consistent with the results in Ref. [22] that one will obtain an entire line of solutions. We have shown data from five different MC runs by coloured circles. The three points near $t_1 \sim 0$ were obtained by starting from a trivial start i.e., $(X, Y = 0)$ while the two points with $t_1 = \pm 3.88(2)$ and $t_2 = 29.41(4)$ were obtained by starting from $X, Y = \pm \mathbb{I}$ respectively. The figure is used after taking permission from the authors of Ref. [22].

bootstrap methods.

## 3.4 Closed and open chain models with three and four matrices

The matrix chain is a $p$ matrices model which was first considered in [32]. We will not mention details of the analytical solution here but instead show the results we obtain for both open and closed versions using MC in Fig. 12 and Fig. 13. This model was also studied in the context of $q$-state Potts model in Refs. [33–35]. We define this model as:

$$Z_p(g, c, \kappa) = \int \mathcal{D}M_1 \cdots \mathcal{D}M_p \exp \mathrm{Tr}\left( \sum_{i=1}^{p} -M_i^2 - g M_i^4 + c \sum_{i=1}^{p-1} M_i M_{i+1} + \kappa M_p M_1 \right). \quad (44)$$

Though exact results are available for any $p$ with $\kappa = 0$, not much has been explicitly done for $p > 3$ since algebra becomes rather involved. When the chain is connected ($\kappa \neq 0$), the model is *not solvable* with $p \geq 4$. We use Monte Carlo methods to study the open and closed cases of the model with $p = 3, 4$ with $N = 300$. It would be interesting if this four matrix model can be *bootstrapped* in the coming years. The $p$-matrices MC code to solve these models (well-tested and currently with for $p = 3, 4$) is available at:

[https://github.com/rgjha/MMMC/3_4MC.py](https://github.com/rgjha/MMMC/3_4MC.py)

It is left as an exercise for the reader to extend this for any $p$. In order to study the model with four matrices and periodic (closed) boundary conditions as defined in (44), we can simply modify `NMAT = 3` to `NMAT = 4`. Note that setting $\kappa = 0$ reduces to the open chain case.

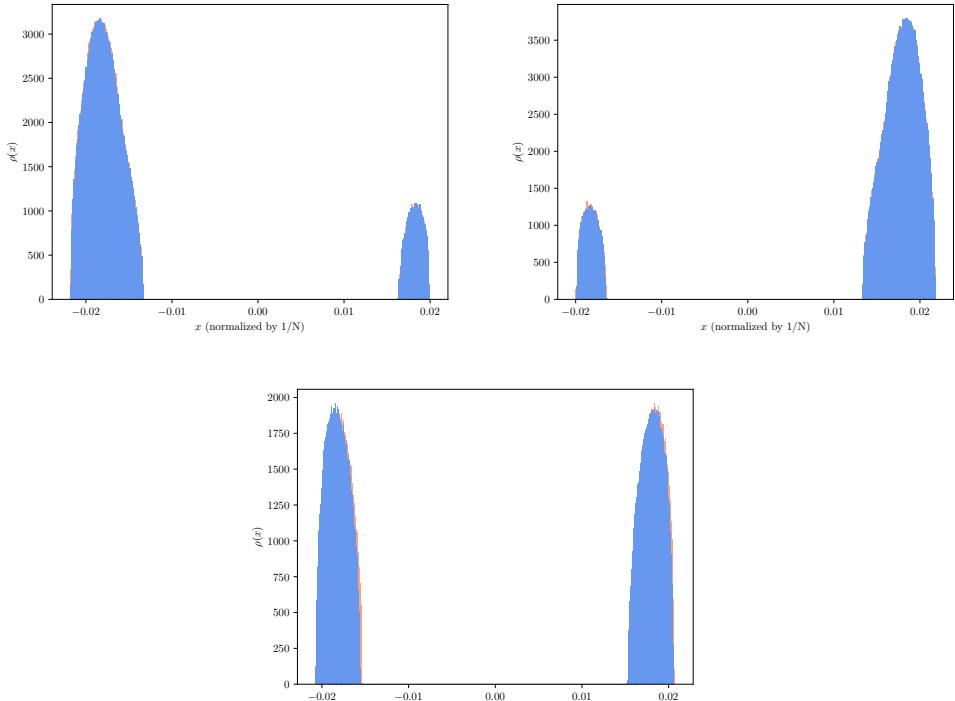

Figure 11: The eigenvalue distribution of two matrices for $g = 1/30$ and $h = 1/15$ with $N = 300$ for the potential given in (43) and shown in Fig. 10. These correspond to green (top left), magenta (top right), red (bottom) data points respectively of Fig. 10. Note that blue and orange distribution overlaps and means that we have $X \to Y$ symmetry as expected.

The broken symmetry correspond to $c \neq \kappa$ but we have not considered it here. The results for $p = 4$ closed symmetric case is given in Table 2.

Table 2: The results obtained for closed chain model with four matrices with $g = 1, 2$ and $N = 300$ at fixed $c = \kappa = 1.35$.

| $g$ | $t_2$ | $t_4$ |
|---|---|---|
| 1 | 1.577(2) | 3.093(3) |
| 2 | 0.741(2) | 0.825(3) |

## 3.5 Models with $D$ matrices including mass terms

After our discussion on models involving one and two matrices, we now turn to matrix models with three or more matrices which we take to be Hermitian. The model is defined as:

$$Z = \int \mathcal{D}X_1 \cdots \mathcal{D}X_D \ \exp\left[ -Nh \sum_i \mathrm{Tr}X_i^2 + \frac{N\lambda}{4} \sum_{i<j} \mathrm{Tr}[X_i, X_j]^2 \right]. \tag{45}$$

If we consider $X_i \mapsto (1 + \epsilon)X_i$, it must leave $Z$ invariant, one arrives at the following exact relation:

$$D(N^2 - 1) = 2h\langle \mathrm{Tr}X_i^2 \rangle - N\lambda\langle \mathrm{Tr}[X_i, X_j]^2 \rangle. \tag{46}$$

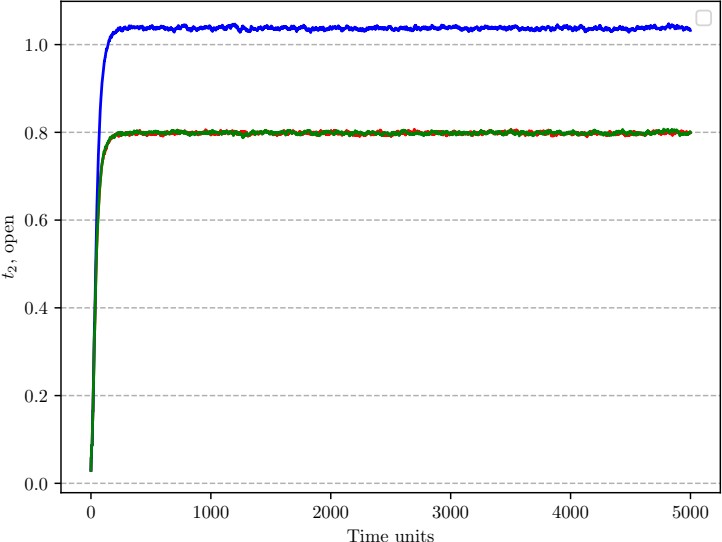

Figure 12: We find that two matrices have common $t_2 = 0.798(3)$ and the third has $t_2 = 1.037(3)$. This is for $g = 1, c = 1.35, \kappa = 0$ and $N = 300$ for the model with three matrices. We have computed errors after discarding the first 1000 time units in this case.

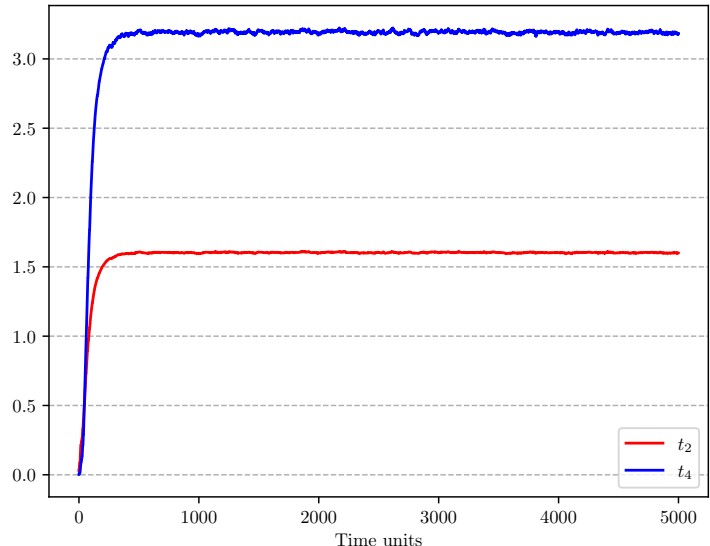

Figure 13: We find that for closed model with three matrices we find $t_2 = 1.603(5)$ and $t_4 = 3.193(5)$. This is for $g = 1, c = \kappa = 1.35$ for $N = 300$. We have computed errors after discarding the first 1000 time units and using jackknife blocking. Note that for this set of parameters it seems like $t_2 = t_4/2$. We found that for $g = 2, c = \kappa = 1.35, t_2 = 0.775(2)$ and $t_4 = 0.887(3)$. It is straightforward to understand the behaviour as a function of $g$ at fixed $c, \kappa$ if that is of interest to the reader by using the codes we provide.

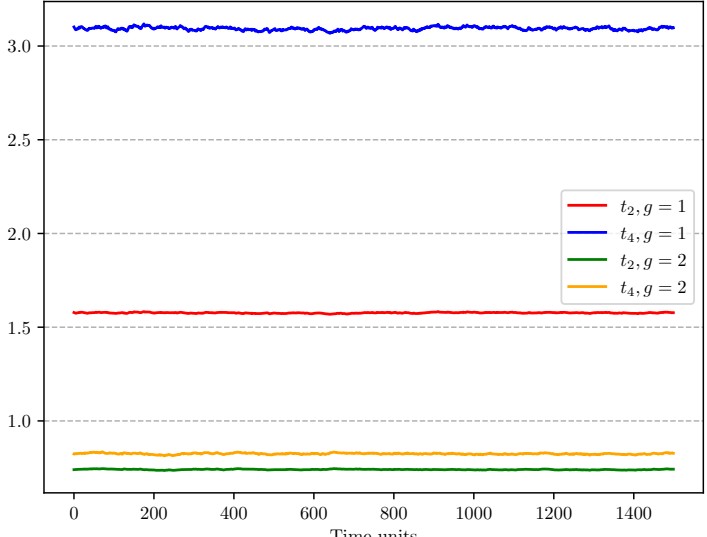

Figure 14: We show the expectation value for $N = 300, c = \kappa = 1.35$ for two different $g$ for closed chain case with four matrices. These runs have not started from a `trivial` start which is understood by noting that the traces are non-zero at the first time unit. See Appendix C for details. The data is given in Table 2

This serves as a check of the MC code and is satisfied to a very good accuracy after ignoring the thermalization cut. We studied this model for $D = 3, 5$ with $h = 1, \lambda = 4$ and compute:

$$R^2 = \frac{1}{DN}\left\langle \operatorname{Tr}\sum_{i=1}^{D} X_i^2 \right\rangle, \quad R^4 = \frac{1}{DN}\left\langle \operatorname{Tr}\sum_{i=1}^{D} X_i^4 \right\rangle. \tag{47}$$

The results are given in Table 3. We note that it is easy to get the sign of $\mathcal{O}(1/N)$ corrections using Monte Carlo methods. The simplest way is to do another set of MC evolution at lower $N$ and see how $R^2$ and $R^4$ change. It is an interesting problem (in practice) to understand how one can apply bootstrap methods away from the planar limit where factorization no longer holds. If this can be done, then it will be very interesting to compare finite $N$ MC results with bootstrap in the future.

Table 3: The results obtained for $D = 3, 5$ matrices models with mass terms are given for $\lambda = 4, h = 1$ with $N = 300$.

| $D$ | $R^2$ | $R^4$ |
|---|---|---|
| 3 | 0.279(4) | 0.158(5) |
| 5 | 0.212(3) | 0.091(5) |

• Exercise 8: Study the model defined by (45) for $D = 3$ by modifying the code given in Appendix E for studying the Yang-Mills type model defined by (48). Check that results are consistent with Table 3.

## 3.6  Multi-matrix Yang-Mills models

In previous Sec. 3.5, we discussed the generalization of Hoppe type matrix models to $D$ matrices with mass terms. It is also interesting to consider these models without mass terms i.e., $h = 0$ in (45) with $D$ matrices. We refer to these models as 'Yang-Mills' type models following Refs. [36, 37] and refer the reader to these for more details. The action is given by:

$$S = -\frac{N}{4\lambda} \int \text{Tr}\left( \sum_{i<j} [X_i, X_j]^2 \right), \qquad (48)$$

where $i, j = 1 \cdots D$. The MC code to study this model is available at:

https://github.com/rgjha/MMMC/YMtype.py

This model is just the general version of the well-known bosonic part of the IKKT model where $D = 10$. But, this model can be studied for general $D$ and has interesting features, see Ref. [38]. Shortly after the BFSS matrix model[10] was proposed as a description of M-theory, the authors of [53] considered a reduction of the quantum-mechanical model (now called IKKT) down to zero dimensions and conjectured it to describe the Type IIB superstrings. Though a complete large $N$ solution of even this model is out of reach, there are a lot of numerical results available which were all inspired by the seminal work of applying Monte Carlo approach to M-theory [54]. There has been some recent work which takes the master-field approach to the IKKT model [55] and is a promising direction but it is not fully clear how well it works. The action of IKKT model is schematically written as:

$$S = \frac{N}{4\lambda} \int \text{Tr}\left( \frac{1}{4} [X_\mu, X_\nu]^2 + \overline{\psi} \Gamma^\mu [X_\mu, \psi] \right), \qquad (49)$$

where $X_\mu$ and $\psi$ are $N \times N$ Hermitian matrices and $\psi$ is ten-dimensional Majorana-Weyl spinor field and the indices run from $1 \cdots D$ with $D = 10$. This model in zero dimensions possesses no usual space-time supersymmetry and is the dimensional reduction of $\mathcal{N} = 1$ super Yang-Mills (SYM) theory in ten dimensions. It is expected that in this model both space and time should be generated from the dynamics of large matrices. This model has no free parameters since $\lambda$ can be absorbed in the field redefinitions. It is also possible to consider variants of this model where $D < 10$. One might worry whether the partition function is convergent at all because of the integral measure being over non-compact $X$. These convergence issues of the partition function of these models for different $D$ were studied by Refs. [36, 37] and we refer the reader to those for additional details. In what follows, we will ignore the fermionic term and only focus on the commutator/bosonic term. One of the observables (also known as 'size' or i.e., the extent of scalars) which we compute in these models is already defined in (47). It is known that $R^2$ behaves as $\sqrt{\lambda}$ in the large $N$ limit as discussed in Ref. [38] but we have not found any previous work which computes the coefficient. In supersymmetric matrix models like BFSS/BMN, this extent of scalars has a dual interpretation in terms of the radius of the dual black hole horizon topology. Our numerical results suggests that $R^2 = 0.361(2)\sqrt{\lambda}$ and $R^4 = 0.266(3)\lambda$ with $N = 300$ for a wide range of couplings i.e., $\lambda \in [1, 100]$. We also note

---

[10]This matrix model was proposed in Ref. [39] and the proposal related the uncompactified eleven-dimensional $M$-theory in light cone frame to the planar limit of the supersymmetric matrix quantum mechanics describing $D0$-branes. This model has been well-studied using MC methods [40–46] though the first explorations of these used a Gaussian approximation method [47, 48] The publicly available code by different groups to study these models, their mass deformation (BMN matrix model) [49] and higher-dimensional systems which describes $D1/D2$ branes [50–52] is available at https://github.com/daschaich/susy, while the highly efficient code for only BFSS and BMN models is available at https://sites.google.com/site/hanadamasanori/home/mmmm. The discussion of these models is not the goal of this article.

that we find the $\sqrt{\lambda}$ and $\lambda$ behaviour for $R^2$ and $R^4$ valid down to $D = 3$. One of the standard tests we do for the reliability of our numerical results is computing the average action. It can be shown that under a change: $X \to e^\epsilon X$, if we demand that $Z$ is invariant, then we find:

$$\frac{\langle S \rangle}{N^2 - 1} = \frac{D}{4}. \tag{50}$$

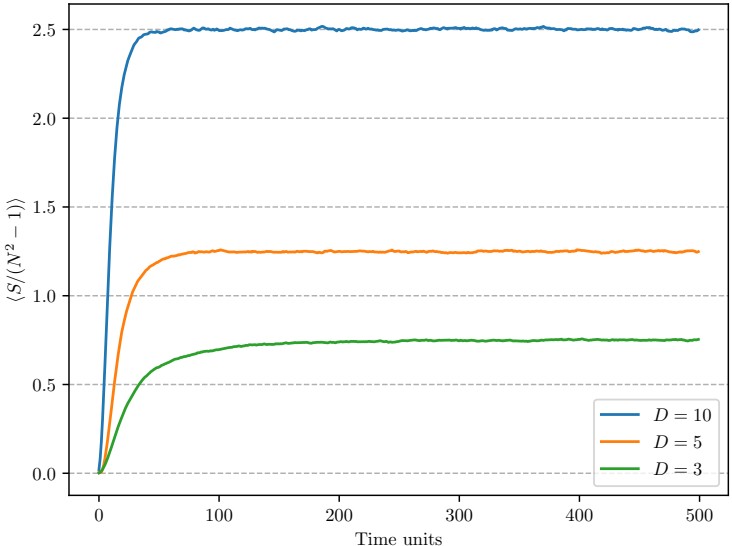

Figure 15: The average action (normalized) for the YM type model with various $D$.

> • Exercise 9: Derive (50) by doing the change: $X_\mu \mapsto e^\epsilon X_\mu$ in (48) and ignoring $\mathcal{O}(\epsilon^2)$ terms.

This is an exact result and must be satisfied during the evolution. We show in Fig. 15 that for $D = 3, 5, 10$ we get this expected result and hence the MC results are correct. The timings for generating 500 time units or trajectories with $N = 300$ is about `50000,2300,2000` seconds for $D = 10, 5, 3$ respectively on a 2.4 GHz i5 A1989 MacBook Pro. The results are collected in Table 4.

Table 4: The results obtained for various $D$ matrices YM models are given for $\lambda = 1$ with $N = 300$.

| $D$ | $R^2$ | $R^4$ |
| --- | --- | --- |
| 3 | 1.129(3) | 2.71(2) |
| 5 | 0.608(2) | 0.765(3) |
| 10 | 0.361(2) | 0.266(3) |

> • Exercise 10: Carry out the MC computation for the Yang-Mills type matrix model with $D = 6$. After sufficient thermalization cut, check that the result for average action is consistent with that obtained using Schwinger-Dyson equations within errors. Refer to

Appendix E for the PYTHON code.

## 4 Summary and future directions

We have described the Monte Carlo method to study different matrix models in the large N limit starting with the simplest one-matrix Hermitian matrix model and then considering models with two and three matrices before carrying on to Yang-Mills type models with up to ten matrices. We obtained new results and confirmed several known results from analytical and bootstrap methods. Though matrix models play a very important role in different areas of Physics, most of them are analytically not solvable, and resorting to numerical techniques also turn up only a handful of methods with their own merits and problems. The recent progress in bootstrap methods looks promising and one can hope that there might be interesting ways to combine these two numerical methods and understand the matrix models in more detail. The Monte Carlo approach to matrix models discussed in this review plays a fundamental role in the first-principle verification of holographic dualities as explained in the main text. Though we have restricted mostly to zero-dimensional models, there are a lot of numerical results for the matrix quantum mechanics in the literature. One of the future goals of these numerical methods is to obtain the master field configuration, as proposed more than forty years ago [56]. Another promising direction that will certainly be explored in the coming years is the application of hybrid quantum algorithms in the NISQ era to study these models [57, 58]. We hope this introduction will encourage interested readers to carry out these numerical computations using the programs provided, and will eventually lead to new ways of solving matrix models and to bootstrap models not yet explored.

## Acknowledgements

The author is indebted to Pedro Vieira for discussions, encouragement, and comments on the draft. We thank Vladimir Kazakov and Zechuan Zheng for helpful email correspondence and for allowing us to use a figure from their paper. We also thank Nikhil Kalyanapuram for general discussions. The author is supported by a postdoctoral fellowship at the Perimeter Institute for Theoretical Physics. Research at Perimeter Institute is supported in part by the Government of Canada through the Department of Innovation, Science and Economic Development Canada and by the Province of Ontario through the Ministry of Economic Development, Job Creation and Trade.

## A Ising model on a random graph

We mentioned in Sec. 2.2 that the method of polynomials can be used to solve the Ising model on random graph. One can show that the partition function written in the usual form:

$$Z = \int \prod d\lambda_i \Delta^2(\lambda) e^{-N \sum V(\lambda_i)}, \tag{51}$$

can be written as (see Ref. [7] for details):

$$Z = N! \, a_0^N \prod_{k=1}^{N-1} f_k^{N-k}, \tag{52}$$

Table 5: Summary of critical exponents obtained for two-dimensional Ising model.

| Crit. exponents | Ising model on random planar graph | Ising model on regular lattice |
|---|---|---|
| $\alpha$ | $-1$ | $0$ |
| $\beta$ | $1/2$ | $1/8$ |
| $\gamma$ | $2$ | $7/4$ |
| $\delta$ | $5$ | $15$ |
| $\nu d$ | $3$ | $2$ |
| $\gamma_{\text{str}}$ | $-1/3$ | $-$ |

where $f_k := a_k/a_{k-1}$. Hence, solving the matrix model is mapped to an equivalent problem of solving for the normalizations appearing in (18). This method was also used to study the Ising Model on a random graph as two-matrix model [59] where the partition function is given by:

$$Z = \int \mathcal{D}A\mathcal{D}B \exp N \text{Tr}\left( -A^2 - B^2 + 2cAB - g\frac{A^3}{3} - g\frac{B^3}{3} \right). \tag{53}$$

Note that this has a $\mathbb{Z}_2$ symmetry because of the partition function being invariant under $A \mapsto B$. This is however broken in finite magnetic fields ($h \neq 0$) and the partition fuction in this case is given by:

$$Z = \int \mathcal{D}A\mathcal{D}B \exp N \text{Tr}\left( -A^2 - B^2 + 2cAB - g_A e^h\frac{A^3}{3} - g_B e^{-h}\frac{B^3}{3} \right). \tag{54}$$

Soon after the solution of the Ising model on a random graph, this was extended to admit magnetic fields [60] as well. We will not discuss the entire solution but will sketch the solution. In this paper, the authors also computed the critical exponents and found different results than Onsager's case for a regular square lattice. The exponents satisfied the usual Essam-Fisher and Rushbrooke's identity ($\alpha + 2\beta + \gamma = 2$) first suggested in Ref. [61] followed after a few months in Ref. [62] and the Widom's scaling law: $\gamma/\beta = \delta - 1$. These values coincide with the exponents obtained for a three-dimensional spherical model. This is a striking correspondence between exponents of two different models in different dimensions! In fact, after few years, while discussing the Yang-Lee edge singularity on a dynamical graph, it was shown that an additional exponent $\sigma = 1/2$ also behaved accordingly. We have listed the exponents in Table (5) for the interested reader. The solution proceeds as follows. We start by rewriting the partition function in terms of eigenvalues as:

$$Z = \int dX dY \Delta(X)\Delta(Y) \exp\left[ -N \sum_i (x_i^2 + y_i^2 + 2cx_iy_i + 4ge^h x_i^4 + 4ge^{-h} y_i^4) \right]. \tag{55}$$

It is now clear that we would need two polynomials $P_k(x)$ and $Q_j(y)$ for this case such that their determinant matches $\Delta(X)$ and $\Delta(Y)$ respectively. These polynomials satisfy the following orthonormal condition:

$$\int dx dy e^{-NV(x,y)} P_k(x)Q_j(y) = h_k \delta_{kj}. \tag{56}$$

They also satisfy several recursion relations for which the interested reader can refer to [60]:

$$Z = \int dX dY \det[P_r(x_k)] \det[Q_r(y_k)] \exp\left[ -N \sum V(X,Y) \right], \tag{57}$$

where we have denoted $\sum_i (x_i^2 + y_i^2 + 2cx_iy_i + 4ge^h x_i^4 + 4ge^{-h}y_i^4)$ by $V(X,Y)$. Transforming to the eigenvalue basis of both matrices $X$ and $Y$ and using the expansion of the determinant we get:

$$
\begin{aligned}
Z &= \epsilon^{i_1 \cdots i_N} \epsilon^{j_1 \cdots j_N} \int dx_{1 \cdots N} dy_{1 \cdots N} P_{i_1}(x_1) \cdots P_{i_N}(x_N) Q_{j_1}(y_1) \cdots Q_{j_N}(y_N) e^{-N \sum V(x_i, y_i)} \\
&= \epsilon^{i_1 \cdots i_N} \epsilon^{j_1 \cdots j_N} \prod_{r=1}^{N} \int dx_r dy_r e^{-NV(x_r, y_r)} P_{i_r}(x_r) Q_{j_r}(y_r) \\
&= N! \prod_{i=0}^{N-1} h_i .
\end{aligned}
\tag{58}
$$

We can define $f_k := h_k/h_{k-1}$ and hence (58) implies:

$$
\log Z_N(c,g,h) = \log N! + N \log h_0 + \sum_{k=1}^{N-1} (N-k) \log f_k .
\tag{59}
$$

One is usually interested in computing the quantity (the subscript 'pc' denotes planar/continuum limit i.e., $N \to \infty$):

$$
F_{pc} = \frac{1}{N^2} \log \left( \frac{Z(c,g,h)}{Z(c,0,0)} \right) = \frac{1}{N} \sum_{k=1}^{N-1} \left( 1 - \frac{k}{N} \log \left( \frac{f_k}{f_{k,0}} \right) \right) .
\tag{60}
$$

## B  MATHEMATICA code for one-matrix model solution

We now give the details to solve the one matrix model in MATHEMATICA. For this we consider the partition function where the potential is given by:

$$
V(Y) = \frac{Y^2}{2} + \frac{gY^4}{4} .
$$

Then we follow the standard procedure described in Sec. 2 and move to the eigenvalue basis and take the $N \to \infty$ limit and define $V(y)$ which is potential in terms of eigenvalues of Y. As we have discussed in the main text, for this case, the higher moments of the trace of $Y$ are related to the second moment and hence we will just calculate $\mathrm{Tr}\, Y^2$ (normalized by $N$) in the planar limit. We give the code below for computing $t_2$ with a fixed $g = 1$. The reader is encouraged to try and change $g$ and see how the results change.

```
V[y_]=y^2/2+(g y^4)/4;
G[x_]=Integrate[-1/(2\[Pi]I)Sqrt[x^2-a^2]/Sqrt[y^2-a^2](N
    V'[y])/(x-y),{y,-a,a},Assumptions->{x>a,a>0}];
sol=Series[G[x],{x, \[Infinity], 1}]-N/x//Simplify//Solve[# ==
    0,a]&//Simplify;
Series[G[x],{x,\[Infinity], 5}]//Normal;
{Coefficient[\%, x, -3]}/N;
\% /. sol;
\%/.{g -> 1}//Chop//N//Grid
```

## C  Brief instructions and explanations to use the PYTHON code

We provide programs which can be used to study several different types of Hermitian matrix models. The instructions on how to use them with run time parameters can be found at:

<https://github.com/rgjha/MMMC/README>

The list of all programs which are available are also summarized below:

1. One matrix model: <https://github.com/rgjha/MMMC/1MM.py>

2. Two matrix Hoppe-type models: <https://github.com/rgjha/MMMC/2MM.py>

3. Three and four matrices chain models: <https://github.com/rgjha/MMMC/3_4MMC.py>

4. Yang-Mills type models: <https://github.com/rgjha/MMMC/YMtype.py>

In addition to the online access of the codes, we also give the programs for one-matrix model and *D* matrices Yang-Mills models in Appendix D, and E respectively. These codes can also be modified to study other potentials as required. In general, only two sub-routines need to be changed. The first is `def potential(X)` and other is the `def force(X)`. The first involves (mostly) trace of product of matrices and the second is the derivative of those matrix traces. These codes in PYTHON are rather short and are all individual programs are less than $300$ lines with lot of common parts like: Leapfrog integrator, Metropolis step, generating random matrices, saving/reading configuration file, and plotting the data. This precise presentation and the choice of programming language are motivated by the fact that often pure theorists and non-MC practitioners think that MC is some magic or is rather difficult to implement[11] but this is certainly not true. Our goal is to explain and use MC for matrix models in the simplest possible manner such that it can be understood by anyone who even remotely wants to understand it. We have not tried to do any optimizations to make the programs more efficient and the only motivation is that someone who has never run a MC code can do so quickly and use it as guidance for deriving exact results or for bootstrapping purposes. In order to run the code correctly, we need to take care of a few things which we discuss below.

- The acceptance rate should always be more than 50% on an average. If the acceptance rate is less than this, we must reduce the size of the time step in the leapfrog integrator by reducing `dt` in the global definitions at the starting. Note that reducing this time step makes the computation more expensive. The non-conservation of energy (`delta H`) in the codes should scale with $(dt)^2$ for a well-defined range of $dt$. This time step might also need to be modified accordingly if we want to explore values of *N* more than $N = 300$ which we mostly use in this article. The code will give a warning if the acceptance falls below 50%. We also should not change the step size $dt$ during the entire evolution of the system. It has to be chosen to a value where acceptance is reasonable and then kept constant. This is related to the fact that changing it will create a bias in sampling which is not desired. If the acceptance doesn't improve even after reducing the time step, it signals an error in the potential or the force terms.

- As a thumb rule, during the evolution, `delta H` (which is the sum of trace of potential (or action) and momenta) should fluctuate around zero with both negative and positive signs. However, this might not be true from the start, and should be monitored after sufficient thermalization. After thermalization, we should have $\langle e^{-\Delta H} \rangle = 1$ within errors. If this is consistently violated and is more than $5\sigma$ away from 1, it means there is a bug. It is straightforward to prove this and can be found in Appendix G.

- We usually start a run by setting all matrices to zero (also referred here as `fresh` or `trivial` start). Then as the evolution progresses, we store a new configuration by

---

[11]I had a similar mindset when I started writing my first MC program for Witten's supersymmetric quantum-mechanical model. But, in few weeks, this disappeared. I am grateful to Simon Catterall for this exercise.

rewriting the older one every 10 time units. The configuration file stores $N \times N$ matrices over which we do the matrix integral in binary format as a `numpy array`. The size of this file can vary from a few MB up to 50 MB or more depending on `NMAT` and `N` (which we call `NCOL` in the code). If we are not doing the run for the first time, it is better to read-in the configuration file since this will save the thermalization time as it will pick up from where it left last time. Note that this can only be done if `NMAT` and `NC` are the same or else it will throw an error. The user can modify these choices easily to suit their requirements.

- The code produces output files ending with `.txt` and `.txt` which are moments of the matrices. The number of columns in these files will be equal to the number of different matrices we considered in the potential i.e., `NMAT`. If we consider a matrix model with ten matrices (the maximum we have used in this article), these files will have ten corresponding columns. It might however be true that they are all very similar to each other if the problem has some specific symmetry such as the one we discussed in 3.3.

- It is common practice among those who use Monte Carlo to never measure an observable every time unit (because of autocorrelation[12], see Sec. 3.1.3). To do this, we can ask the program to save data (moments of matrices) only after some fixed number of time steps. This is controlled by `GAP` in all the codes. Another way to make sure the data is not correlated is use a sufficiently large block size when computing errors using the jackknife method.

- Monte Carlo is based on a sampling method and will always have errors for the expectation values. The errors must be carefully computed using either jackknife binning or some other method. Usually, the errors go down as $\sim 1/\sqrt{N_{\text{data}}}$ up to autocorrelation effects as one accumulates more data.

Though these codes have been checked many times and compared to known solutions (where available), it is possible that there might still be minor bugs in them. If you encounter a problem or have questions, please contact the author.

# D  PYTHON code for Hermitian one matrix model

We provide the code in this section to study the 1MM using the Monte Carlo method. By running the code given below on a modern laptop, we get the result shown in Fig. 6. We can readily extend this code (by changing `NMAT`) to study matrix models where the integration is over several different matrices. In order to run this code using Mac/Linux system (assuming PYTHON is installed with required libraries) we can just type the following in a terminal from the directory with the program: `python 1MM.py 0 1 300 500`. The code takes four input arguments. The first two are binary arguments related to whether we are reading any old configuration file and whether we want to save the one which will be generated, `0 1`, means that we are not reading any configuration but we want to save it for later use. The third argument is the matrix size. Ideally, planar limit is $N \rightarrow \infty$ but here we have `N = 300`. The last argument is the number of trajectories for which we want to run the program. For this model, we found that to converge about `500` should be enough but to accurately get several digits of accuracy, more than `5000` maybe needed. It takes about 40 minutes to run 500 time units with `N = 300` on a modern laptop.

---

[12]See `https://emcee.readthedocs.io/en/stable/user/autocorr/` for details

```python
#!/usr/bin/python3
# -*- coding: utf-8 -*-
import time
import datetime
import sys
import numpy as np
import random
import math
import os
from numpy import linalg as LA
from matplotlib.pyplot import *
from matplotlib import pyplot as plt

startTime = time.time()
print ("STARTED:" , datetime.datetime.now().strftime("%d %B %Y, %H:%M:%S"))
if len(sys.argv) < 5:
  print("Usage:python",str(sys.argv[0]),"READ-IN? " "SAVE-or-NOT? " "NCOL "
      "NITERS")
  sys.exit(1)

READIN = int(sys.argv[1])
SAVE = int(sys.argv[2])
NCOL = int(sys.argv[3])
Niters_sim = int(sys.argv[4])

NMAT = 1
g = 1.0
dt = 1e-3
nsteps = int(0.5/dt)
GAP = 2.
cut = int(0.25*Niters_sim)

if Niters_sim%GAP != 0:
  print("'Niters_sim' mod 'GAP' is not zero ")
  sys.exit(1)
if READIN not in [0,1]:
    print ("Wrong input for READIN")
    sys.exit(1)
if SAVE not in [0,1]:
    print ("Wrong input for SAVE")
    sys.exit(1)

X = np.zeros((NMAT, NCOL, NCOL), dtype=complex)
mom_X = np.zeros((NMAT, NCOL, NCOL), dtype=complex)
f_X = np.zeros((NMAT, NCOL, NCOL), dtype=complex)
X_bak = np.zeros((NMAT, NCOL, NCOL), dtype=complex)
HAM, expDH, trX2, trX4, MOM = [], [], [], [], []
t2_ex = [None] * NMAT
t4_ex = [None] * NMAT

print ("Matrix integral simulation of%2.0f MM"%(NMAT))
print ("NCOL =" " %3.0f " "," " and g =" " %4.2f" % (NCOL, g))
print ("--------------------------------------------------")

def dagger(a):
  return np.transpose(a).conj()

def box_muller():
  PI = 2.0*math.asin(1.0);
  r = random.uniform(0,1)
  s = random.uniform(0,1)
  p = np.sqrt(-2.0*np.log(r)) * math.sin(2.0*PI*s)
  q = np.sqrt(-2.0*np.log(r)) * math.cos(2.0*PI*s)
  return p,q

def copy_fields(b):
  for j in range(NMAT):
    X_bak[j] = b[j]
  return X_bak
```

```python
def rejected_go_back_old_fields(a):
  for j in range(NMAT):
    X[j] = a[j]
  return X

def refresh_mom():
  for j in range (NMAT):
    mom_X[j] = random_hermitian()
  return mom_X

def random_hermitian():
  tmp = np.zeros((NCOL, NCOL), dtype=complex)
  for i in range (NCOL):
    for j in range (i+1, NCOL):
      r1, r2 = box_muller()
      tmp[i][j] = complex(r1, r2)/math.sqrt(2)
      tmp[j][i] = complex(r1, -r2)/math.sqrt(2)
  for i in range (NCOL):
    r1, r2 = box_muller()
    tmp[i][i] = complex(r1, 0.0)
  return tmp

def makeH(tmp):
  tmp2 = 0.50*(tmp+dagger(tmp)) - \
      (0.50*np.trace(tmp+dagger(tmp))*np.eye(NCOL))/NCOL
  for i in range (NCOL):
    tmp2[i][i] = complex(tmp[i][i].real,0.0)
  if np.allclose(tmp2, dagger(tmp2)) == False:
    print ("WARNING: Couldn't make hermitian")
  return tmp2

def hamil(X,mom_X):
  ham = potential(X)
  for j in range (NMAT):
    ham += 0.50 * np.trace(np.dot(mom_X[j],mom_X[j])).real
  return ham

def potential(X):
  pot = 0.0
  for i in range (NMAT):
    pot += 0.50 * np.trace(np.dot(X[i],X[i])).real
    pot += (g/4.0)* np.trace(X[i] @ X[i] @ X[i] @ X[i]).real
  return pot*NCOL

def force(X):
  for i in range (NMAT):
    f_X[i] = (X[i] + (g*(X[i] @ X[i] @ X[i])))*NCOL
  for j in range(NMAT):
    if np.allclose(f_X[j], dagger(f_X[j])) == False:
      f_X[j] = makeH(f_X[j])
  return f_X

def leapfrog(X,dt):

  mom_X = refresh_mom()
  ham_init = hamil(X,mom_X)

  for j in range(NMAT):
    X[j] += mom_X[j] * dt * 0.5

  for i in range(1, nsteps+1):
    f_X = force(X)
    for j in range(NMAT):
      mom_X[j] -= f_X[j] * dt
      X[j] += mom_X[j] * dt

  f_X = force(X)
  for j in range(NMAT):
    mom_X[j] -= f_X[j] * dt
    X[j] += mom_X[j] * dt * 0.5
```

```python
    ham_final = hamil(X,mom_X)
    return X, ham_init, ham_final

def update(X, acc_count):

    X_bak = copy_fields(X)
    X, start, end = leapfrog(X, dt)
    change = end - start
    expDH.append(np.exp(-1.0*change))
    if np.exp(-change) < random.uniform(0,1):
        X = rejected_go_back_old_fields(X_bak)
        print(("REJECT: deltaH = " "%10.7f" " " "startH = " "%10.7f" " endH = "
            "%10.7f" % (change, start, end)))
    else:
        print(("ACCEPT: deltaH = " "%10.7f" " "startH = " "%10.7f" " endH = "
            "%10.7f" % (change, start, end)))
        acc_count += 1

    if MDTU%GAP == 0:
        t2_ex[0] = np.trace(np.dot(X[0],X[0])).real
        trX2.append(t2_ex[0]/NCOL)
        t4_ex[0] = np.trace((X[0] @ X[0] @ X[0] @ X[0])).real
        trX4.append(t4_ex[0]/NCOL)

        if NMAT > 1:
            for i in range (1, NMAT):
                t2_ex[i] = np.trace(np.dot(X[i],X[i])).real
                t4_ex[i] = np.trace((X[i] @ X[i] @ X[i] @ X[i])).real

        for item in t2_ex:
            f3.write("%4.8f " % (item/NCOL))
        for item in t4_ex:
            f4.write("%4.8f " % (item/NCOL))
        f3.write("\n")
        f4.write("\n")

    return X, acc_count

if __name__ == '__main__':

    if READIN == 0:
        print ("Loading fresh configuration")
        for i in range (NMAT):
            X[i] = 0.0

    if READIN == 1:

        name_f = "config_1MM_N{}.npy".format(NCOL)
        if os.path.isfile(name_f) == True:
            print ("Reading old configuration file:", name_f)
            A = np.load(name_f)
            for i in range (NMAT):
                for j in range (NCOL):
                    for k in range (NCOL):
                        X[i][j][k] = A[i][j][k]
            for j in range(NMAT):
                if np.allclose(X[j], dagger(X[j])) == False:
                    print ("Input configuration 'X' not hermitian, ", LA.norm(X[j] -
                        dagger(X[j])), "making it so")
                    X[j] = makeH(X[j])
        else:
            print ("Configuration not found, loaded fresh")
            for i in range (NMAT):
                X[i] = 0.0

    f3 = open('t2_1MM_N%s_g%s.txt' %(NCOL,round(g,4)), 'w')
    f4 = open('t4_1MM_N%s_g%s.txt' %(NCOL,round(g,4)), 'w')
```

```python
acc_count = 0.
for MDTU in range (1, Niters_sim+1):

    X, acc_count = update(X, acc_count)
    if MDTU%10 == 0 and SAVE == 1:
        name_f = "config_1MM_N{}.npy".format(NCOL)
        print ("Saving configuration file: ", name_f)
        np.save(name_f, X)

f3.close()
f4.close()

if NMAT == 1:
    t2_exact = (((12*g)+1)**(3/2.) - 18*g - 1)/(54*g*g)
    # Exact result for 1MM quartic potential with g = 1
    plt.rc('text', usetex=True)
    plt.rc('font', family='serif')
    MDTU = np.linspace(0, int(Niters_sim/GAP), int(Niters_sim/GAP),
        endpoint=True)
    plt.ylabel(r'Tr(X$^2)/N$',fontsize=12)
    plt.xlabel('Time units', fontsize=12)
    plt.grid(which='major', axis='y', linestyle='--')
    plt.axhline(y=t2_exact, color='teal', linestyle='--')
    plt.figure(1)
    plot (MDTU, trX2, 'teal')
    outname = "1MM_N%s_g%s" %(NCOL, g)
    plt.savefig(outname+'.pdf')
print ("-------------------------------------------------------")
print ("Acceptance rate: ", (acc_count/Niters_sim)*100,"%")
if acc_count/Niters_sim < 0.50:
    print("WARNING: Acceptance rate is below 50%")

if READIN == 0:
    trX2 = trX2[cut:]
    trX4 = trX4[cut:]
    expDH = expDH[cut:]

print ("COMPLETED:" , datetime.datetime.now().strftime("%d %B %Y,
    %H:%M:%S"))
endTime = time.time()
print ("Running time:", round(endTime - startTime, 2), "seconds")
```

# E Python code for YM matrix model with $D$ matrices

In this section, we provide the MC code which can be used to study a YM matrix model with $D$ matrices and especially $D = 10$ which is the bosonic sector of the well-known IKKT model. More details about the model and its relation to the nonperturbative formulations of string theory can be found in Ref. [38]. It is left as a simple exercise for the reader to compare the differences between this program and the one matrix model code given before in Appendix D.

```python
#!/usr/bin/python
# -*- coding: utf-8 -*-
import numpy as np
from numpy import linalg as LA
from numpy.linalg import matrix_power
import time
import os
import datetime
import sys
import random
import math
import scipy as sp
```

```python
import scipy.linalg
from scipy.linalg import expm
from matplotlib.pyplot import *
from matplotlib import pyplot as plt
from matplotlib.backends.backend_pdf import PdfPages
from matplotlib import pyplot

startTime = time.time()
print ("STARTED:" , datetime.datetime.now().strftime("%d %B %Y %H:%M:%S"))

if len(sys.argv) < 7:
  print("Usage: python", str(sys.argv[0]), "READ-IN? " "SAVE-or-NOT? " "NCOL"
      " "NITERS " "D " "LAMBDA ")
  sys.exit(1)

READIN = int(sys.argv[1])
SAVE = int(sys.argv[2])
NCOL = int(sys.argv[3])
Niters_sim = int(sys.argv[4])
NMAT = int(sys.argv[5])
LAMBDA = float(sys.argv[6])
if NMAT < 2:
    print ("NMAT must be at least two")
    sys.exit(1)
if READIN not in [0,1]:
    print ("Wrong input for READIN")
    sys.exit(1)
if SAVE not in [0,1]:
    print ("Wrong input for SAVE")
    sys.exit(1)

COUPLING = float(NCOL/(4.0*LAMBDA))
GENS = NCOL**2 - 1
dt = 5e-4
nsteps = int(1e-2/dt)
GAP = 1
t2 = np.zeros((NMAT),dtype=float)
t4 = np.zeros((NMAT),dtype=float)
X = np.zeros((NMAT, NCOL, NCOL), dtype=complex)
mom_X = np.zeros((NMAT, NCOL, NCOL), dtype=complex)
f_X = np.zeros((NMAT, NCOL, NCOL), dtype=complex)
X_bak = np.zeros((NMAT, NCOL, NCOL), dtype=complex)
HAM, expDH, ACT, scalar = [],[],[],[]

print ("Yang-Mills type matrix model with %2.0f matrices" % (NMAT))
print ("NCOL = " "%3.0f " "," " and coupling = " " %4.2f" % (NCOL, COUPLING))
print ("-------------------------------------------")

def dagger(a):
    return np.transpose(a).conj()

def box_muller():
    PI = 2.0*math.asin(1.0);
    r = random.uniform(0,1)
    s = random.uniform(0,1)
    p = np.sqrt(-2.0*np.log(r)) * math.sin(2.0*PI*s)
    q = np.sqrt(-2.0*np.log(r)) * math.cos(2.0*PI*s)
    return p,q

def comm(A,B):
    return np.dot(A,B) - np.dot(B,A)

def unit_matrix():
    matrix = np.zeros((NCOL, NCOL), dtype=complex)
    for i in range (NCOL):
        matrix[i][i] = complex(1.0,0.0)
    return matrix

def copy_fields(b):
    for j in range(NMAT):
```

```python
        X_bak[j] = b[j]
    return X_bak

def rejected_go_back_old_fields(a):
    for j in range(NMAT):
        X[j] = a[j]
    return X

def refresh_mom():
    for j in range (NMAT):
        mom_X[j] = random_hermitian()
    return mom_X

def random_hermitian():
    tmp = np.zeros((NCOL, NCOL), dtype=complex)
    for i in range (NCOL):

        for j in range (i+1, NCOL):
            r1, r2 = box_muller()
            tmp[i][j] = complex(r1, r2)/math.sqrt(2)
            tmp[j][i] = complex(r1, -r2)/math.sqrt(2)

    for i in range (NCOL):
        r1, r2 = box_muller()
        tmp[i][i] = complex(r1, 0.0)
    return tmp

def makeH(tmp):
    tmp2 = 0.50*(tmp+dagger(tmp)) -
        (0.50*np.trace(tmp+dagger(tmp))*np.eye(NCOL))/NCOL
    for i in range (NCOL):
        tmp2[i][i] = complex(tmp[i][i].real,0.0)
    if np.allclose(tmp2, dagger(tmp2)) == False:
        print ("WARNING: Couldn't make hermitian.")
    return tmp2

def hamil(mom_X):
    s = 0.0
    for j in range (NMAT):
        s += 0.50 * np.trace(np.dot(dagger(mom_X[j]),mom_X[j])).real
    return s

def potential(X):
    s1 = 0.0
    for i in range (NMAT):
        for j in range (i+1, NMAT):
            co = np.dot(X[i],X[j]) - np.dot(X[j],X[i])
            tr = np.trace(np.dot(co,co))
            s1 -= COUPLING*tr.real
    return s1

def force(X):

    tmp_X = np.zeros((NMAT, NCOL, NCOL), dtype=complex)
    for i in range (NMAT):
        for j in range (NMAT):
            if i == j:
                continue
            else:
                temp = comm(X[i], X[j])
                tmp_X[i] -= comm(X[j], temp)
        f_X[i] = 2.0*COUPLING*dagger(tmp_X[i])

    for j in range(NMAT):
        if np.allclose(f_X[j], dagger(f_X[j])) == False:
            f_X[j] = makeH(f_X[j])
    return f_X

def leapfrog(X,mom_X, dt):
    for j in range(NMAT):
```

```python
            X[j] += mom_X[j] * dt/2.0
    f_X = force(X)

    for step in range(nsteps):
        for j in range(NMAT):
            mom_X[j] -= f_X[j] * dt
            X[j] += mom_X[j] * dt
        f_X = force(X)

    for j in range(NMAT):
        mom_X[j] -= f_X[j] * dt
        X[j] += mom_X[j] * dt/2.0

    return X, mom_X, f_X

def update(X):
    mom_X = refresh_mom()
    s1 = hamil(mom_X)
    s2 = potential(X)
    start_act = s1 + s2
    X_bak = copy_fields(X)
    X, mom_X, f_X = leapfrog(X,mom_X,dt)
    s1 = hamil(mom_X)
    s2 = potential(X)
    end_act = s1 + s2
    change = end_act - start_act
    HAM.append(abs(change))
    expDH.append(np.exp(-1.0*change))

    if np.exp(-change) < random.uniform(0,1):
        X = rejected_go_back_old_fields(X_bak)
        print(("REJECT: deltaH = " "%10.7f" " " startH = " "%10.7f" " endH = "
            "%10.7f" % (change, start_act, end_act)))
    else:
        print(("ACCEPT: deltaH = " "%10.7f" "startH = " "%10.7f" " endH = "
            "%10.7f" % (change, start_act, end_act)))

    ACT.append(s2)
    tmp = 0.0
    for i in range (0,NMAT):
        val = np.trace(X[i] @ X[i]).real/NCOL
        val2 = np.trace(X[i] @ X[i] @ X[i] @ X[i]).real/NCOL
        t2[i] = val
        t4[i] = val2
        tmp += val

    tmp /= NMAT
    scalar.append(tmp)
    if MDTU%GAP == 0:
        f3.write("%4.8f \n" % (s2/GENS))
        for item in t2:
            f4.write("%4.8f " % item)
        for item in t4:
            f5.write("%4.8f " % item)
        f4.write("\n")
        f5.write("\n")

    return X

if __name__ == '__main__':

    if READIN == 0:
        print ("Starting from fresh")
        for i in range (NMAT):
            X[i] = 0.0

    if READIN == 1:
        name_f = "config_YM_N{}_l_{}_D_{}.npy".format(NCOL, LAMBDA, NMAT)
```

```python
    if os.path.isfile(name_f) == True:
        print ("Reading old configuration file:", name_f)

        A = np.load(name_f)
        for i in range (NMAT):
            for j in range (NCOL):
                for k in range (NCOL):
                    X[i][j][k] = A[i][j][k]

        for j in range(NMAT):
            if np.allclose(X[j], dagger(X[j])) == False:
                print ("Input configuration not hermitian, making it so")
                X[j] = makeH(X[j])
    else:
        print ("Can't find config. file for this NCOL and LAM")
        print ("Starting from fresh")
        for i in range (NMAT):
            X[i] = 0.0

f3 = open('action_N%s_D%s.txt' %(NCOL,NMAT), 'w')
f4 = open('t2_N%s_D%s.txt' %(NCOL,NMAT), 'w')
f5 = open('t4_N%s_D%s.txt' %(NCOL,NMAT), 'w')

for MDTU in range (1, Niters_sim+1):
    X = update(X)

    if MDTU%10 == 0 and SAVE == 1:
        name_f = "config_YM_N{}_l_{}_D_{}.npy".format(NCOL, LAMBDA, NMAT)
        print ("Saving configuration file: ", name_f)
        np.save(name_f, X)

ACT = [x/GENS for x in ACT]
f3.close()
f4.close()
f5.close()
print ("-------------------------------------------")
print("<S> = ", np.mean(ACT), "+/-", (np.std(ACT)/np.sqrt(np.size(ACT) -
    1.0)))
print ("COMPLETED:" , datetime.datetime.now().strftime("%d %B %Y
    %H:%M:%S"))
endTime = time.time()
# Plot results!
t2plot = plt.figure(1)
plt.rc('text', usetex=True)
plt.rc('font', family='serif')
MDTU = np.linspace(0, int(Niters_sim/GAP), int(Niters_sim/GAP),
    endpoint=True)
plt.ylabel(r'$\langle R^2 \rangle$')
plt.xlabel('Time units')
plot(MDTU, scalar, 'teal')
plt.grid(which='major', axis='y', linestyle='--')
act_plot = plt.figure(2)
plt.ylabel(r'$\langle S/(N^2-1) \rangle$')
plt.xlabel('Time units')
plt.axhline(y = NMAT/4.0, color='blue', linestyle='--')
plot(MDTU, ACT, 'blue')
plt.grid(which='major', axis='y', linestyle='--')
outname = "YM_N%s_D%s" %(NCOL,NMAT)
pp = PdfPages(outname+'.pdf')
pp.savefig(t2plot, dpi = 300, transparent = True)
pp.savefig(act_plot, dpi = 300, transparent = True)
pp.close()
print ("Running time:", round(endTime - startTime, 2), "seconds")
```

# F   Computing error using Jackknife method

We give a sample code for computing statistical errors of output data files in PYTHON and refer the reader to Ref. [63] for more details. The program can be used from the terminal as follows: `python jk_error.py t2.txt 200 20 0`. This means that we ask the code to take out the first 200 time units data for thermalization cut and we set the size of the block to be 20. The last argument tells the program which column to consider for averaging with 0 meaning the first column. To ensure that we have used a reasonable thermalization cut, one can check for different cuts and see if the results are same within errors. One can do the same for the block size.

```python
#!/usr/bin/python3
import sys
import numpy as np
import itertools
from math import *
data = []; data_tot = 0. ; Data = [] ; data_jack = []

if len( sys.argv ) > 4:
    filename = sys.argv[1]
    therm_cut = int(sys.argv[2])
    blocksize = int(sys.argv[3])
    which_column = int(sys.argv[4])

if len( sys.argv ) <= 4:
    print("NEED 4 ARGUMENTS : FILE THERM-CUT BLOCKSIZE COLUMN_TO_PARSE")
    sys.exit()

file = open(filename, "r")
for line in itertools.islice(file, therm_cut, None):

    line = line.split()
    if which_column > int(np.shape(line)[0])-1:
        print ("Column to average does not exist")
        sys.exit(1)
    data_i = float(line[which_column])
    data.append(data_i)
    data_tot += data_i
    n = len(data)

n_b = int(n/blocksize)
B = 0.

for k in range(n_b):
    for w in range((k*blocksize)+1,(k*blocksize)+blocksize+1):
        B += data[w-1]
    Data.insert(k,B)
    B = 0

''' Do the jackknife estimates '''
for i in range(n_b-1):
    data_jack.append((data_tot - Data[i]) / (n - blocksize))
    data_av = data_tot / n # Do the overall averages
    data_av = data_av
    data_jack_av = 0.; data_jack_err = 0.
for i in range(n_b-1):
    dR = data_jack[i]
    data_jack_av += dR
    data_jack_err += dR**2

data_jack_av /= n_b-1
data_jack_err /= n_b-1

data_jack_err = sqrt((n_b - 2) * abs(data_jack_err - data_jack_av**2))
print(" %8.7f " " %6.7f" " %6.2f" % (data_jack_av, data_jack_err, n_b))
```

# G   Solutions to selected Exercises

⋆ Solution to Exercise 2:

We now show that $\det(V) = \prod_{i<j}(\lambda_i - \lambda_j)$ where $V$ is:

$$
V = \begin{pmatrix}
1 & \lambda_1 & \lambda_1^2 & \cdots & \lambda_1^{N-1} \\
1 & \lambda_2 & \lambda_2^2 & \cdots & \lambda_2^{N-1} \\
\vdots & \vdots & \ddots & \vdots & \vdots \\
1 & \lambda_N & \lambda_N^2 & \cdots & \lambda_N^{N-1}
\end{pmatrix} = \lambda_i^{j-1}
$$

We first note that the determinant is unchanged if we make the change to all columns except the first given by:

$$
\lambda_i^{j-1} \to \lambda_i^{j-1} - \lambda_1 \lambda_i^{j-2}\,, \tag{61}
$$

then we have,

$$
\det(V') = \begin{vmatrix}
1 & 0 & 0 & \cdots & 0 \\
1 & \lambda_2 - \lambda_1 & \lambda_2(\lambda_2 - \lambda_1) & \cdots & \lambda_2^{N-2}(\lambda_2 - \lambda_1) \\
\vdots & \vdots & \ddots & \vdots & \vdots \\
1 & \lambda_N - \lambda_1 & \lambda_N(\lambda_N - \lambda_1) & \cdots & \lambda_N^{N-2}(\lambda_N - \lambda_1)
\end{vmatrix}. \tag{62}
$$

By using Laplace Expansion formula for determinants, along the first row we find that $\det(V') = \det(V'')$ where $V''$ is:

$$
\det(V'') = \begin{vmatrix}
\lambda_2 - \lambda_1 & \cdots & \cdots & \lambda_2^{N-2}(\lambda_2 - \lambda_1) \\
\vdots & \vdots & \ddots & \vdots \\
\lambda_N - \lambda_1 & & \cdots & \lambda_N^{N-2}(\lambda_N - \lambda_1)
\end{vmatrix}. \tag{63}
$$

Taking the factors common in each row, we get:

$$
\det(V) = \det(V'') = (\lambda_2 - \lambda_1)\cdots(\lambda_N - \lambda_1)\begin{vmatrix}
1 & \cdots & \cdots & \lambda_2^{N-2} \\
\vdots & \vdots & \ddots & \vdots \\
1 & & \cdots & \lambda_N^{N-2}
\end{vmatrix}. \tag{64}
$$

If we iterate this with the smaller matrix, it is easy to see that we obtain:

$$
\det(V) = \prod_{i<j}(\lambda_j - \lambda_i). \tag{65}
$$

To define a Vandermonde matrix and compute determinant of a $5 \times 5$ matrix, we can execute following command in MATHEMATICA:

```
V = Table[Subscript[\[Alpha], i]^j, {i, 1, 5}, {j, 0, 4}];
Det@V // Simplify
```

⋆ Solution to Exercise 3 and additional comments:

The basic idea of the loop equations of matrix models is to capture the invariance of the model under field redefinitions. This is also sometimes known as 'Schwinger-Dyson (SD) equations'. One of the exercises in the main text is to derive these equations. Here, we give the proof for

the interested reader. We start by noting that the integral of the total derivative vanish and hence:

$$\sum_{i,j} \int dM \frac{\partial}{\partial M_{ij}} \left( (M^k)_{ij} \, e^{-N \text{Tr} V(M)} \right) = 0, \tag{66}$$

By computing the derivatives and using large $N$ factorization, we obtain:

$$\left\langle \text{Tr} M^k V'(M) \right\rangle = \sum_{l=0}^{k-1} \langle \text{Tr} M^l \rangle \langle \text{Tr} M^{k-l-1} \rangle . \tag{67}$$

In the steps above, we have used two identities:

$$\frac{\partial}{\partial M_{ij}} (M^k)_{ij} = \sum_{l=0}^{k-1} (M^l)_{ii} (M^{k-l-1})_{jj}, \tag{68}$$

and,

$$\frac{\partial}{\partial M_{ij}} e^{-N \text{Tr} V(M)} = -N V'(M)_{ji} \, e^{-N \text{Tr} V(M)}, \tag{69}$$

where $V'$ denotes the derivative with respect to the matrix. However, these loop equations are not valid when the integration is over some other matrix ensembles (such as orthogonal/symplectic). It is better to start from the eigenvalue integral representation to consider general $\beta$. We can write moments as:

$$\langle \text{Tr} M^k \rangle = \frac{1}{Z} \int \Delta(\lambda)^\beta d\lambda_1 \cdots d\lambda_N \, \exp\left( -\frac{N\beta}{2} \sum_i V(\lambda_i) \right) \sum_{i=1}^{N} \lambda_i^k . \tag{70}$$

It is easy to derive 'generalized loop equations' from here using the fact that integral of total derivative vanishes. We obtain:

$$\left\langle \text{Tr} M^k V'(M) \right\rangle + \underbrace{k \left( \frac{2}{\beta} - 1 \right) \text{Tr} M^{k-1}}_{\text{zero for } \beta = 2} = \sum_{l=0}^{k-1} \langle \text{Tr} M^l \rangle \langle \text{Tr} M^{k-l-1} \rangle . \tag{71}$$

⋆ Solution to Exercise 4:

Considering (17) with $k = 1$ and quartic potential, we get:

$$\left\langle \text{Tr} \left( M^2 + g M^4 \right) \right\rangle = 1. \tag{72}$$

This then implies,

$$\frac{1}{N} \text{Tr} M^4 \equiv t_4 = \frac{1 - t_2}{g}. \tag{73}$$

We can extend this to $\text{Tr} M^6 / N \equiv t_6$ which can be obtained in terms of $t_2$ as:

$$t_6 = \frac{2 t_2 - \frac{(1 - t_2)}{g}}{g}. \tag{74}$$

⋆ Solution to Exercise 5:

We now show that $\langle e^{-\Delta H} \rangle = 1$ when phase-space are is preserved under evolution. The Hamiltonian of the system is defined as:

$$H(P,X) = \frac{1}{2}\mathrm{Tr}P^2 + N\mathrm{Tr}V(X), \tag{75}$$

where $X$ will be set of matrices involved in the model. If we assume that the area of phase space is preserved under evolution i.e., $dPdX = dP'dX'$, we then have:

$$Z = \int dP'dX'e^{-H'}$$
$$= \int dPdXe^{-H}e^{H-H'}, \tag{76}$$

Dividing (76) by $Z$ we get,

$$\langle e^{H-H'} \rangle = \langle e^{-\Delta H} \rangle = 1. \tag{77}$$

⋆ Solution to Exercise 8:

We need to modify the potential and the corresponding forces as discussed in Appendix C. In addition to the `def potential(X)` and `def force(X)` given in Appendix E, we add mass terms to potential and corresponding forces as sketched below:

```
def potential(X):
    for i in range (NSCALAR):
    massterm += h * NCOL * np.trace(X[i] @ X[i]).real

def force(X):
    for i in range (NSCALAR):
        f_X[i] += 2.0 * h * NCOL * X[i]
```

⋆ Solution to Exercise 9:

We consider $X_\mu \rightarrow (1+\epsilon)X_\mu + \mathcal{O}(\epsilon^2)$ and $\mathcal{D}X \rightarrow (1 + \epsilon D(N^2-1))\mathcal{D}X$ with the path integral:

$$Z = \int \mathcal{D}Xe^{-S} = \int \mathcal{D}X \exp\left[-\frac{1}{4g^2}\mathrm{Tr}[X_\mu, X_\nu]^2\right]. \tag{78}$$

The transformation changes $Z$ by:

$$Z = Z + \epsilon\left\{D(N^2-1)Z - 4\langle S\rangle Z\right\} = Z(1 + \epsilon\left\{D(N^2-1) - 4\langle S\rangle\right\}). \tag{79}$$

If we demand that $Z$ remains invariant, term in the parenthesis should vanish and we get the desired result:

$$\frac{D}{4} = \frac{\langle S\rangle}{N^2-1}. \tag{80}$$

⋆ Solution to the Exercise in Footnote 3:

Executing following command in MATHEMATICA will check that Wigner distribution is observed. The deviation from the semi-circle distribution can be seen for small $n$.[13]

---

[13]Please see https://www.wolfram.com/language/11/random-matrices for further details

```
n = 1000;
scaledSpectrum=Flatten[RandomVariate[scaledSpectrum\[ScriptCapitalD][n],
    100]];
Show[Histogram[scaledSpectrum, {0.05}, PDF, ChartStyle -> LightOrange],
Plot[PDF[WignerSemicircleDistribution[1], x],{x, -1.5, 1.5}, PlotLegends ->
    None,
PlotStyle -> ColorData[27, 1]], ImageSize -> Medium]"
```

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
