# Peer review of "Introduction to Monte Carlo for Matrix Models"

_SciPost Physics Lecture Notes, doi:SciPost Phys. Lect. Notes 46 (2022)_

## Round 1 · Referee Report · Anonymous (Referee 1) · 2022-1-25

Strengths

The lecture notes provide a clear and gentle introduction into numerical methods for large N matrix models. It incorporates also nicely some very recent developments concerning a bootstrap approach to matrix models. It can sometimes be hard for someone interested in this sort of material to get into it as it requires a lot of programming. The lecture notes remove this drawback as they come with the codes and a short accompanying explanation. This allows the researcher to immediately understand the physics she/he is ultimately interested in.

Weaknesses

This is not really a weakness, but personally I always like to have a short section in a lecture note about the 'way forward' or 'current issues'. This would explain a little bit what the methods that are explained in the lecture notes are currently used for and what the obstacles are. This might spark the reader into going beyond the things discussed in the lecture notes.

Report

-

---

## Round 1 · Referee Report · Anonymous (Referee 2) · 2022-2-19

Report

I think this review and sample programs are useful materials for students and researchers interested in numerical approaches to matrix models. However, I would like to suggest a few points which can be improved.

Sec.3.1 seems to be a little bit out of place. Matrix bootstrap is a numerical method, but not the main topic of this article (Monte Carlo methods). Perhaps it can be moved to Sec.2. (Then the title of Sec.2 has to be modified as well.)

The HMC algorithm is explained in Sec.3.2. Although the algorithm is clearly explained, laypeople may wonder why the right ensemble is obtained by using this algorithm. Probably it is better to introduce the Metropolis algorithm as the simplest example of the Markov Chain Monte Carlo methods and explain the basic logic behind such algorithms. Then, as a byproduct, the advantage of the HMC algorithm can be illuminated.
(A question which is somewhat related to this part:
The HMC algorithm is particularly useful when dynamical fermions are involved. For the bosonic matrix models, is it possible to use the heat-bath algorithm?)

In Sec.3.2.1, it is better to introduce the word "Box-Muller algorithm". (This name is used in sample code.)

Reference [39] did not study the D0-brane quantum mechanics. Better references are hep-th/0803.4273 and hep-th/0707.4454. Another reference that achieved good numerical precision is 1503.08499[hep-lat].
In this context, it would be good to refer to hep-th/9910001 and hep-th/hep-th/0007051. They used the Gaussian approximation method to study D0-brane quantum mechanics numerically and demonstrated the power of the numerical approach to holography.

In Appendix G, would it be possible to show the solutions to all exercises?

Requested changes

Please find the suggestions in the report.

---

## Round 2 · Author Response

List of changes
- We have merged Sec.3.1 with Sec.2 and renamed the section.
- We have added 1-2 paragraphs regarding the Metropolis-Hastings algorithm as the simplest example of MCMC (Markov chain Monte Carlo).
- For the bosonic fields, it is certainly possible to use heat-bath, but the article introduces HMC since it is more natural when advancing to field theories with fermions where usually RHMC is used (rational HMC). We have made a comment about this in the article.
- We have introduced “Box-Muller algorithm” in Sec.3.2.1.
- We have corrected the fact that Ref. [39] did not study D0 model and added the four references.
- We have added future directions as suggested by the first referee and added references.

---

## Round 2 · List of Changes

- We have merged Sec.3.1 with Sec.2 and renamed the section.
- We have added 1-2 paragraphs regarding the Metropolis-Hastings algorithm as the simplest example of MCMC (Markov chain Monte Carlo).
- For the bosonic fields, it is certainly possible to use heat-bath, but the article introduces HMC since it is more natural when advancing to field theories with fermions where usually RHMC is used (rational HMC). We have made a comment about this in the article.
- We have introduced “Box-Muller algorithm” in Sec.3.2.1.
- We have corrected the fact that Ref. [39] did not study D0 model and added the four references.
- We have added future directions as suggested by the first referee and added references.

---

## Editorial Decision

published